# Hyper-Representations as Generative Models: Sampling Unseen Neural Network Weights

**Konstantin Schürholt**
konstantin.schuerholt@unisg.ch
AIML Lab, School of Computer Science
University of St.Gallen

**Boris Knyazev**
b.knyazev@samsung.com
Samsung - SAIT AI Lab, Montreal

**Xavier Giró-i-Nieto**
xavier.giro@upc.edu
Institut de Robòtica i Informàtica Industrial
Universitat Politècnica de Catalunya

**Damian Borth**
damian.borth@unisg.ch
AIML Lab, School of Computer Science
University of St.Gallen

## Abstract

Learning representations of neural network weights given a model zoo is an emerging and challenging area with many potential applications from model inspection, to neural architecture search or knowledge distillation. Recently, an autoencoder trained on a model zoo was able to learn a *hyper-representation*, which captures intrinsic and extrinsic properties of the models in the zoo. In this work, we extend hyper-representations for generative use to sample new model weights. We propose layer-wise loss normalization which we demonstrate is key to generate high-performing models and several sampling methods based on the topology of hyper-representations. The models generated using our methods are diverse, performant and capable to outperform strong baselines as evaluated on several downstream tasks: initialization, ensemble sampling and transfer learning. Our results indicate the potential of knowledge aggregation from model zoos to new models via hyper-representations thereby paving the avenue for novel research directions.

## 1 Introduction

Over the last decade, countless neural network models have been trained and uploaded to different model hubs. Many factors such as random initialization and no global optimum ensure that the trained models are different from one another. What could we learn from such a population of neural network models? Since the parameter space of neural networks is complex and high-dimensional, representation learning from such populations (often referred to as model zoos) has become an emerging and challenging area.

Recent work along that direction has demonstrated the ability of such learned representations to capture intrinsic and extrinsic properties of the models in a zoo [40, 37, 27]. According to [37], NNs populate a low dimensional manifold, which can be learned with an autoencoder via self-supervised learning directly from the model paramters (weights and biases) without access to the original image data and labels. This so called *hyper-representation* has been demonstrated to be useful to predict several model properties such as accuracy, hyperparameters or architecture configurations.

However, [37] focused on discriminative downstream tasks by exploiting the encoder only. We take one step further and extend their work towards the generative downstream tasks by sampling model weights directly from the task-agnostic hyper-representation. To that end, we introduce a layer-wise normalization that improves the quality of decoded neural network weights significantly. Based on

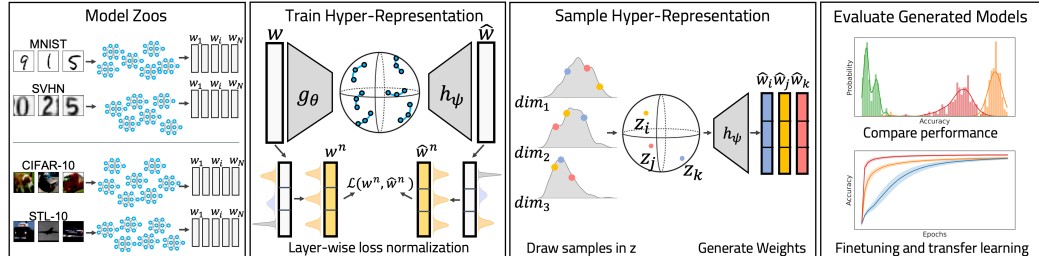

Figure 1: Outline of our approach: Model zoos are trained on image classification tasks. Hyper-representations are trained with self-supervised learning on the weights of the model zoos using layer-wise loss normalization in the reconstruction loss. We sample new embeddings in hyper-representation space and decode to weights. Generated models perform significantly better than random initialization or models sampled from baseline hyper-representations. Sampled models achieve high performance fine-tuned and transfer learned on new datasets.

a careful analysis of the geometry, smoothness and robustness of this space, we also propose several sampling methods to generate weights in a single forward pass from the hyper-representation. We evaluate our approach on four image datasets and three generative downstream tasks of (i) model initialization, (ii) ensemble sampling, and (iii) transfer learning. Our results demonstrate its capability to out-perform previous hyper-representation learning and conventional baselines.

Previous work on generating model weights proposed (Graph) HyperNetworks [14, 45, 21], Bayesian HyperNetworks [8], HyperGANs [35] and HyperTransformers [46] for neural architecture search, model compression, ensembling, transfer- or meta-learning. These methods learn representations by using images and labels of the target domain. In contrast, our approach only uses model weights and does not need access to underlying data samples and labels – an emergent use case, e.g. of deep learning monitoring services or model hubs. In addition to the ability to generate novel and diverse model weights, compared to previous works our approach (a) can generate novel weights conditionally on model zoos from unseen tasks and (b) can be conditioned on the latent factors of the underlying hyper-representation. Notably, both (a) and (b) can be done without the need to retrain hyper-representations.

The results suggest our approach (Figure 1) to be a promising step towards a general purpose hyper-representation encapsulating knowledge of model zoos to advance different downstream tasks. The hyper-representations and code to reproduce our results are available at https://github.com/HSG-AIML/NeurIPS_2022-Generative_Hyper_Representations.

## 2   Background: Training Hyper-Representations

We summarize the first stage of our method that corresponds to learning a hyper-representation of a population of neural networks, called a *model zoo* [37]. In [37] and this paper, a model zoo consists of models trained on the same task such as CIFAR-10 image classification [23]. Specifically, a hyper-representation is learned using an autoencoder $\hat{\mathbf{w}}_i = h(g(\mathbf{w}_i))$ on a zoo of $M$ models $\{\mathbf{w}_i\}_1^M$, where $\mathbf{w}_i$ is the flattened vector of dimension $N$ of all the weights of the $i$-th model. The encoder $g$ compresses vector $\mathbf{w}_i$ to fixed-size hyper-representation $\mathbf{z}_i = g(\mathbf{w}_i)$ of lower dimension. The decoder $h$ decompresses the hyper-representation to the reconstructed vector $\hat{\mathbf{w}}_i$. Both encoder and decoder are built on a self-attention block [41]. The samples from model zoos are understood as sequences of convolutional or fully connected neurons. Each of the neurons is encoded as a token embedding and concatenated to form a sequence. The sequence is passed through several layers of multi-head self-attention. Afterwards, a special compression token summarizing the entire sequence is linearly compressed to the bottleneck. The output is fed through a tanh-activation to achieve a bounded latent space $\mathbf{z}_i$ for the hyper-representation. The decoder is symmetric to the encoder, the embeddings are linearly decompressed from hyper-representations $\mathbf{z}_i$ and position encodings are added.

Training is done in a multi-objective fashion, minimizing the composite loss $\mathcal{L} = \beta\mathcal{L}_{MSE} + (1-\beta)\mathcal{L}_c$, where $\mathcal{L}_c$ is a contrastive loss and $\mathcal{L}_{MSE}$ is a weight reconstruction loss (see details in [37]). We can write the latter in a layer-wise way to facilitate our discussion in § 3.1:

$$\mathcal{L}_{MSE} = \frac{1}{MN} \sum_{i=1}^{M} \sum_{l=1}^{L} ||\hat{\mathbf{w}}_i^{(l)} - \mathbf{w}_i^{(l)}||_2^2, \qquad (1)$$

where $\hat{\mathbf{w}}_i^{(l)}, \mathbf{w}_i^{(l)}$ are reconstructed and original weights for the $l$-th layer of the $i$-th model in the zoo. The contrastive loss $\mathcal{L}_c$ leverages two types of data augmentation at train time to impose structure on the latent space: permutation exploiting inherent symmetries of the weight space and random erasing.

## 3 Methods

In the following, we present (i) layer-wise loss normalization to ensure that decoded models are performant, and (ii) sampling methods to generate diverse populations of models.

### 3.1 Layer-Wise Loss Normalization

We observed that hyper-representations as proposed by [37] decode to dysfunctional models, with performance around random guessing. To alleviate that, we propose a novel layer-wise loss normalization (LWLN), which we motivate and detail in the following.

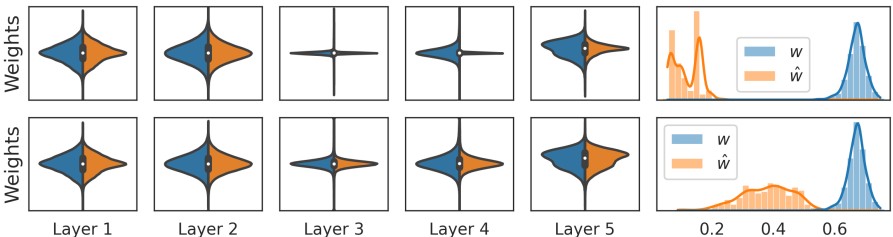

Figure 2: Comparison of the distributions of SVHN zoo weights $\mathbf{w}$ (blue) and reconstructed weights $\hat{\mathbf{w}}$ (orange) as well as their test accuracy on the SVHN test set. **Top:** Baseline hyper-representation as proposed by [37], the weights of layers 3, 4 collapse to the mean. These layers form a weak link in reconstructed models. The accuracy of reconstructed models drops to random guessing. **Bottom:** Hyper-representation trained with layer-wise loss normalization (LWLN). The normalized distributions are balanced, all layers are evenly reconstructed, and the accuracy of reconstructed models is significantly improved.

Due to the MSE training loss in (1), the reconstruction error can generally be expected to be uniformly distributed over all weights and layers of the weight vector $\mathbf{w}$. However, the weight magnitudes of many of our zoos are unevenly distributed across different layers. In these zoos, the even distribution of reconstruction errors lead to undesired effects. Layers with broader distributions and large-magnitude weights are reconstructed well, while layers with narrow distributions and small-magnitude weights are disregarded. The latter layers can become a weak link in the reconstructed models, causing performance to drop significantly down to random guessing. The top row of Figure 2 shows an example of a baseline hyper-representation learned on the zoo of SVHN models [32]. Common initialization schemes [15, 11] produce distributions with different scaling factors per layer, so the issue is not an artifact of the zoos, but can exist in real world model populations. Similarly, recent work on generating models normalizes weights to boost performance [21]. In order to achieve equally accurate reconstruction across the layers, we introduce a layer-wise loss normalization (LWLN) with the mean $\mu_l$ and standard deviation $\sigma_l$ of all weights in layer $l$ estimated over the train split of the zoo:

$$\mathcal{L}_{MSE} = \frac{1}{MN} \sum_{i=1}^{M} \sum_{l=1}^{L} \left\| \frac{\hat{\mathbf{w}}_i^{(l)} - \mu_l}{\sigma_l} - \frac{\mathbf{w}_i^{(l)} - \mu_l}{\sigma_l} \right\|_2^2 = \frac{1}{MN} \sum_{i=1}^{M} \sum_{l=1}^{L} \frac{\|\hat{\mathbf{w}}_i^{(l)} - \mathbf{w}_i^{(l)}\|_2^2}{\sigma_l^2}. \quad (2)$$

### 3.2 Sampling from Hyper-Representations

We introduce methods to draw diverse and high-quality samples $\mathbf{z}^* \sim p(\mathbf{z})$ from the learned hyper-representation space to generate model weights $\mathbf{w}^* = h(\mathbf{z}^*)$. Such sampling is facilitated if there is knowledge on the topology of the space spanned by $\mathbf{z}$. One way to achieve that is to train a variational autoencoder (VAE) with a predefined prior [20] instead of the autoencoder of [37]. While training VAEs on common domains such as images has become well-understood and feasible, in our relatively novel weight domain, we found it problematic (see details in Appendix E). Other generative methods avoid a predefined prior of VAEs, either by analyzing the topology of the space learned by the autoencoder or fitting a separate density estimation model on top of the learned representation [26, 13]. These methods assume the representation space to have strong regularities. The hyper-representation space learned by the autoencoder of [37] is already regularized by dropout regularization applied to the encoder and decoder as in [10]. The contrastive loss component requiring similar models to be embedded close to each other may also improve the regularity of the representation space. Empirically, we found our layer-wise loss normalization (LWLN) to further regularize the representation space by ensuring robustness and smoothness (see Figure 3 in § 4).

Given the smoothness and robustness of the learned hyper-representation space, we follow [26, 13, 10] in estimating the density and topology to draw samples from a regularized autoencoder. To that end, we introduce three strategies to sample from that space: $S_{\text{KDE}}, S_{\text{Neigh}}, S_{\text{GAN}}$. To model the density and topology in representation space, we use the embeddings of the train set as anchor samples $\{\mathbf{z}_i\}$. We observe that many anchor samples from $\{\mathbf{z}_i\}$ correspond to the models with relatively poor accuracy (Figure 2), so to improve the quality of sampled weights, we consider the variants of these methods using only those embeddings of training samples corresponding to the top 30% performing models. We denote these sampling methods as $S_{\text{KDE30}}, S_{\text{Neigh30}}, S_{\text{GAN30}}$ respectively. These methods can potentially decrease sample diversity, however, we found that the generated weights are still diverse enough (e.g. to construct high-performant ensembles, Figure 5). Finally, as baseline and sanity check we explore sampling uniformly in representation space $S_U$ and sampling in low-probability regions $S_C$.

### 3.2.1 Uniform $S_U$

As a naive baseline, we draw samples uniformly in hyper-representation space (bounded by tanh, § 2) and denote it as $S_U$. This is naive, because we found that the embeddings $\mathbf{z}$ populate only sections of a shell of a high-dimensional sphere (see Figures 11 and 12 in Appendix D). So most of the uniform samples lie in the low-probability regions of the space and are not expected to be decoded to useful models.

### 3.2.2 Density estimation $S_{\text{KDE}}$ and counterfactual sampling $S_C$

The dimensionality $D$ of hyper-representations $\mathbf{z}$ in [37], as well as in our work, is relatively high due to the challenge of compressing weights $\mathbf{w}$. Fitting a probability density model to such a high-dimensional distribution is feasible by making a conditional independence assumption: $p(\mathbf{z}^{(j)}|\mathbf{z}^{(k)}, \mathbf{w}) = p(\mathbf{z}^{(j)}|\mathbf{w})$, where $\mathbf{z}^{(j)}$ is the $j$-th dimensionality of the embedding $\mathbf{z}$. To model the distribution of each $j$-th dimensionality, we choose kernel density estimation (KDE), as it is a powerful yet simple, non-parametric and deterministic method with a single hyperparameter. We fit a KDE to the $M$ anchor samples $\{\mathbf{z}_i^{(j)}\}_{i=1}^M$ of each dimension $j$, and draw samples $z^{(j)}$ from that distribution: $z^{(j)} \sim p(\mathbf{z}^{(j)}) = \frac{1}{Mh}\sum_{i=1}^M K\left(\frac{\mathbf{z}^{(j)}-\mathbf{z}_i^{(j)}}{h}\right)$, where $K(x) = (2\pi)^{-1/2}\exp\left(-\frac{x^2}{2}\right)$ is the Gaussian kernel and $h$ is a bandwidth hyperparameter. The samples of each dimension $z^{(j)}$ are concatenated to form samples $\mathbf{z}^* = [z^{(1)}, z^{(2)}, \cdots, z^{(D)}]$. This method is denoted as $S_{\text{KDE}}$.

As a sanity check, we invert the $S_{\text{KDE}}$ method and explicitly draw samples from regions not populated by anchor samples, i.e. with low probability according to the KDE. This method, denoted as $S_C$, essentially samples counterfactual embeddings and similarly to $S_U$ is expected to perform poorly.

### 3.2.3 Neighbor sampling $S_{\text{Neigh}}$

Sampling neighbors of anchor samples $\{\mathbf{z}_i\}$ could be a simple and effective sampling strategy, but due to high sparsity of the hyper-representation space this strategy results in poor-quality samples. We therefore propose to use a neighborhood-based dimensionality reduction function $k : \mathbb{R}^D \to \mathbb{R}^d$ that maps $\mathbf{z}_i$ to low-dimensional embeddings $\mathbf{n}_i \in \mathbb{R}^d$ where sampling is facilitated. The assumption is that due to the low dimensionality of $\mathbb{R}^d$ (we choose $d = 3$) there will be fewer low-probability regions, so that uniform sampling in $\mathbb{R}^d$ can be effective. Specifically, given low-dimensional embeddings $\mathbf{n}_i = k(\mathbf{z}_i)$, we sample $\mathbf{n}^*$ uniformly from the cube: $\mathbf{n}^* \sim U(min(\mathbf{n}), max(\mathbf{n}))$. Samples $\mathbf{n}^*$ are then mapped back to hyper-representations $\mathbf{z}^* = k^{-1}(\mathbf{n}^*)$. To preserve the neighborhood topology of $\mathbb{R}^D$ in $\mathbb{R}^d$ and enable mapping back to $\mathbb{R}^D$, we choose $k$ to be an approximate inverse neighborhood-based dimensionality reduction function based on UMAP [28].

### 3.2.4 Latent space GAN $S_{\text{GAN}}$

A common choice for generative representation learning is generative adversarial networks (GANs) [12]. While training a GAN directly to generate weights is a promising yet challenging avenue for future research [35], we found the GAN framework to work reasonably well when trained on the hyper-representations. This idea follows [26, 13] that showed improved training stability and efficiency compared to training GANs on inputs directly. We train a generator $G : \mathbb{R}^d \to \mathbb{R}^D$ with $\mathbf{z}^* = G(\mathbf{n}^*)$ to generate samples in hyper-representation space from the Gaussian noise $\mathbf{n}^*$. We choose $d = 16$ as a compromise between size and capacity. See a detailed architecture of our GAN in Appendix E.

# 4 Experiments

## 4.1 Experimental Setup

We train and evaluate our approaches on four image classification datasets: MNIST [24], SVHN [32], CIFAR-10 [23], STL-10 [5]. For each dataset, there is a model zoo that we use to train an autoencoder following [37].

**Model zoos:** In practice, there are already many available model zoos, e.g., on Hugging Face or GitHub, that can be used for hyper-representation learning and sampling. Unfortunately, these zoos are not systematically constructed and require further effort to mine and evaluate. Therefore, in order to control the experiment design, ensure feasibility and reproducibility, we generate novel or use the model zoos of [37, 38] created in a systematic way. With controlled experiments, we aim to develop and evaluate inductive biases and methods to train and utilize hyper-representation, which can be scaled up efficiently to large-scale and non-systematically constructed zoos later. For each image dataset, a zoo contains $M = 1000$ convolutional networks of the same architecture with three convolutional layers and two fully-connected layers. Varying only in the random seeds, all models of the zoo are trained for 50 epochs with the same hyperparameters following [37]. To integrate higher diversity in the zoo, initial weights are uniformly sampled from a wider range of values rather than using well-tuned initializations of [11, 15]. Each zoo is split in the train (70%), validation (15%) and test (15%) splits. To incorporate the learning dynamics, we train autoencoders on the models trained for 21-25 epochs following [37]. Here the models have already achieved high performance, but have not fully converged. The development in the remaining epochs of each model are treated as hold-out data to compare against. We use the MNIST and SVHN zoos from [37] and based on them create the CIFAR-10 and STL-10 zoos. Details on the zoos can be found in Appendix A.

**Experimental details:** We train separate hyper-representations on each of the model zoos. Images and labels are not used to train the hyper-representations (see § 2). Using the proposed sampling methods (§ 3.2), we generate new embeddings and decode them to weights. We evaluate sampled populations as initializations (epoch 0) and by fine-tuning for up to 25 epochs. We distinguish between in-dataset and transfer-learning. For in-dataset, the same image dataset is used for training and evaluating our hyper-representations and baselines. For transfer-learning, hyper-representations (and pre-trained models in baselines) are trained on a source dataset, then all populations are evaluated and fine-tuned on a different target dataset. Full details on training, including infrastructure and compute is detailed in the Appendix B.

**Baselines:** As the first baseline, we consider the autoencoder of [37], which is same as ours but without the proposed layer-wise loss-normalization (LWLN, § 3.1). We combine this autoencoder with the $S_{\text{KDE30}}$ sampling method and, hence, denote it as $B_{\text{KDE30}}$. We consider two other baselines based on training models with stochastic gradient descent (SGD): training from scratch on the target classification task $B_T$, and training on a source followed by fine-tuning on the target task $B_F$. The latter remains one of the strongest transfer learning baselines [4, 9, 22].

**Reproducibility, reliability and comparability:** We compare populations of at least 50 models to evaluate each method reliably. We report standard deviation in Tables 1-2 and statistical significance, effect size and 95% confidence interval in Appendix F. To ensure fairness and comparability, all methods share training hyperparameters. Fine-tuning uses the hyperparameters of the target domain.

## 4.2 Results

In the following, we first analyze the learned hyper-representations further justifying our sampling methods and assumptions made in § 3.2. We then confirm the effectiveness of our approach for model initialization without and with fine-tuning in the in-dataset and transfer learning settings.

### 4.2.1 Hyper-Representations are Robust and Smooth

We evaluate the robustness and smoothness of the hyper-representation space with two experiments on the SVHN zoo. First, to evaluate robustness, we add different levels of noise to the embeddings of the test set to create $\tilde{\mathbf{z}}$, decode them to model weights $\tilde{\mathbf{w}}$ and compute models' accuracies on the SVHN classification task. We found that both the baseline as well as our hyper-representations are robust to noise as large levels of relative noise >10% are required to affect performance (Figure 3, a,c). Second, to probe for smoothness, we linearly interpolate between the test set embeddings (i) along

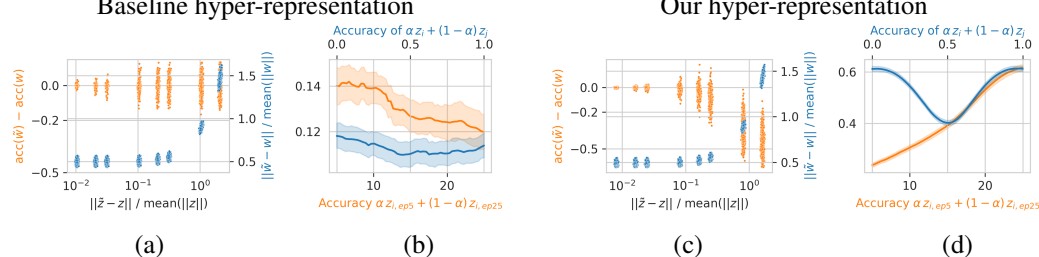

Baseline hyper-representation        Our hyper-representation

(a)        (b)        (c)        (d)

Figure 3: **(a,c):** Robustness of hyper-representations. For both baseline and our hyper-representation, relatively large levels of relative noise >10% are necessary to degrade the test accuracy (orange) or reconstruction (blue); see the text for further discussion. **(b,d):** Interpolations along model trajectories (orange) and between **z** of different models (blue) show the smoothness of our hyper-representation.

the trajectory of the same model at different epochs ($\mathbf{z}_{i,ep5}$ and $\mathbf{z}_{i,ep25}$) and (ii) between 250 random pairs of embeddings on the trajectories of different models ($\mathbf{z}_i$ and $\mathbf{z}_j$). We decode the interpolated embeddings and compute models' accuracies on the classification task. For our model, we found remarkably smooth development of accuracy along the interpolation in both schemes (Figure 3, d). The lack of fluctuations along and between trajectories support both local and global notions of smoothness in hyper-representation space.

For the baseline autoencoder (without LWLN) decoded models all perform close to 10% accuracy, so these representations do not support similar notions of smoothness (Figure 3, b), while robustness can be misleading, since the accuracy even without adding noise is already low (Figure 3, a). Therefore, LWLN together with regularizations added to the autoencoder allow for learning robust and smooth hyper-representation. This property makes sampling from that representation more meaningful as we show next.

### 4.2.2 Sampling for In-dataset Initialization

**Comparison between sampling methods:** We evaluate the performance of different sampled populations (obtained with LWLN) *without fine-tuning* generated weights. On MNIST, all sampled models except those obtained using $S_U$ and $S_C$ perform better than random initialization (10% accuracy), but worse than models trained from

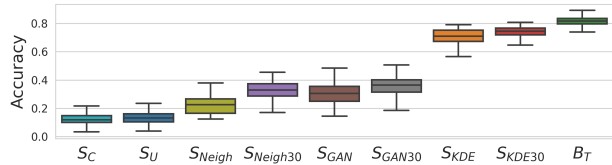

Figure 4: MNIST results of sampled weights (no fine-tuning) compared to training from scratch with SGD ($B_T$).

scratch $B_T$ for 25 epochs (Figure 4). Distribution-based samples ($S_{KDE}$ and $S_{GAN}$) perform better than neighborhood based samples ($S_{Neigh}$). The populations based on the top 30% perform better than their 100% counterparts with $S_{KDE30}$ as the strongest sampling method overall. This demonstrates that the learned hyper-representation and sampling methods are able to capture complex subtleties in weight space differentiating high and low performing models.

**Comparison to the baseline hyper-representations:** We also compare $S_{KDE30}$ that is based on our autoencoder with layer-wise loss normalization (LWLN) to the baseline autoencoder using the same sampling method ($B_{KDE30}$) without fine-tuning. On all datasets except for MNIST, $S_{KDE30}$ considerably outperform $B_{KDE30}$ with the latter performing just above 10% (random guessing), see Table 1 (rows with epoch 0). We attribute the success of LWLN to two main factors. First, LWLN prevents the collapse of reconstruction to the mean (compare Figure 2 top to bottom). Second, by fixing the weak links, the reconstructed models perform significantly better (see Appendix C for more results).

**In-dataset fine-tuning:** When fine-tuning, our $S_{KDE30}$ and baseline $B_{KDE30}$ appear to gradually converge to similar performance (Table 1). While unfortunate, this result aligns well with previous findings that longer training and enough data make initialization less important [30, 17, 34].

We also compare $S_{\text{KDE30}}$ and $B_{\text{KDE30}}$ to training models from scratch ($B_T$). On all four datasets, both ours and the baseline hyper-representations outperform $B_T$ when generated weights are fine-tuned for the same number of epochs as $B_T$. Notably, on MNIST and SVHN generated weights fine-tuned for 25 epochs are even better than $B_T$ run for 50 epochs. Comparison to 50 epochs is more fair though, since the hyper-representations were trained on model weights trained for up to 25 epochs. These findings show that the models initialized with generated weights learn faster achieving better results in 25 epochs than $B_T$ in 50 epochs.

Table 1: Mean and std of test accuracy (%) of sampled populations with LWLN ($S_{\text{KDE30}}$) and without ($B_{\text{KDE30}}$) compared to models trained from scratch $B_T$. Best results for each epoch and dataset are bolded.

| Method | Ep. | MNIST | SVHN | CIFAR-10 | STL-10 |
|---|---|---|---|---|---|
| $B_T$ | 0 | | $\approx$10% (random guessing) | | |
| $B_{\text{KDE30}}$ | 0 | 63.2 ± 7.2 | 10.1 ± 3.2 | 15.5 ± 3.4 | 12.7 ± 3.4 |
| $S_{\text{KDE30}}$ | 0 | **68.6 ± 6.7** | **51.5 ± 5.9** | **26.9 ± 4.9** | **19.7 ± 2.1** |
| $B_T$ | 1 | 20.6 ± 1.6 | 19.4 ± 0.6 | 27.5 ± 2.1 | 15.4 ± 1.8 |
| $B_{\text{KDE30}}$ | 1 | 83.2 ± 1.2 | 67.4 ± 2.0 | 39.7 ± 0.6 | **26.4 ± 1.6** |
| $S_{\text{KDE30}}$ | 1 | **83.7 ± 1.3** | **69.9 ± 1.6** | **44.0 ± 0.5** | 25.9 ± 1.6 |
| $B_T$ | 25 | 83.3 ± 2.6 | 66.7 ± 8.5 | 46.1 ± 1.3 | 35.0 ± 1.3 |
| $B_{\text{KDE30}}$ | 25 | **93.2 ± 0.6** | **75.4 ± 0.9** | 48.1 ± 0.6 | **38.4 ± 0.9** |
| $S_{\text{KDE30}}$ | 25 | 93.0 ± 0.7 | 74.2 ± 1.4 | **48.6 ± 0.5** | 38.1 ± 1.1 |
| $B_T$ | 50 | 91.1 ± 2.6 | 70.7 ± 8.8 | 48.7 ± 1.4 | 39.0 ± 1.0 |

**Sampling ensembles:** We found that a potentially useful by-product of learning hyper-representations is the ability to generate high-performant ensembles at almost no extra computational cost, since both sampling and generation are computationally cheap. To demonstrate this effect, we compare ensembles formed using the baseline autoencoder ($B_{\text{KDE30}}$) and ours ($S_{\text{KDE30}}$) to the ensembles composed of models trained from scratch for 25 epochs ($B_T$) on SVHN. Ensembles generated using the baseline $B_{\text{KDE30}}$ stagnate below 20% (Figure 5). In contrast, ensembles generated using our $S_{\text{KDE30}}$ gracefully improve with the ensemble size outperforming single $B_T$ models and almost matching $B_T$ ensembles with enough models in the ensembles. Remarkably, the average test accuracy of generated ensembles of 15 models is 77.6%, which is considerably higher than 70.7% of models trained on SVHN for 50 epochs. We conclude that hyper-representations learned

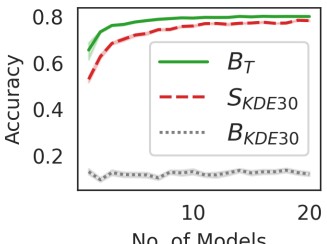

Figure 5: Generated ensembles evaluated on SVHN. Test accuracy is averaged over 15 ensembles of randomly chosen models.

with LWLN generate models that are not only performant, but also diverse. Although generating ensembles requires learning hyper-representation and model zoo first, we assume that in future such a hyper-representation can be trained once and reused in unseen scenarios as we tentatively explore below (see results in Table 3 and the discussion therein).

**Do reconstructed models become similar to the original during fine-tuning?** Sampled hyper-representations often learn faster and to a higher performance than the population of models they were trained on (Table 1). We therefore explore the question, if reconstructed models develop in weight space in the same direction as their original, or find a different solution. On SVHN, we found that the reconstructed models ($\hat{\mathbf{w}}$) after one epoch of fine-tuning perform similar to their originals ($\mathbf{w}$) and slightly outperform from there on (Figure 6, left). At the same

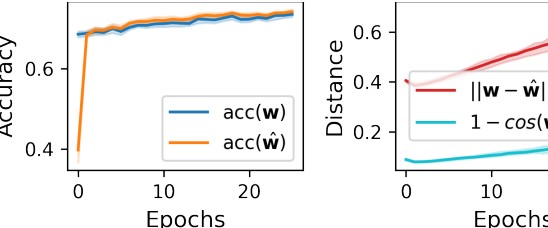

Figure 6: Progression of test accuracy (left) and distance (right) between weights during fine-tuning on SVHN; $\mathbf{w}$ – initialization with the weights trained using SGD for 25 epochs; $\hat{\mathbf{w}}$ – initialization with reconstructed weights.

time, pairs of original and reconstructed models move further apart and become less aligned in weight space (Figure 6, right). It appears that reconstructed models perform better and explore different solutions in weight space to do so. This confirms the intuition that hyper-representations impress useful structure on decoded weights. A pass through encoder and decoder thus results not just in a noisy reconstruction of the original sample. Instead, it maps to a different region on the loss surface, which leads to faster learning and better solutions. Combining this with the ensembling results in Figure 5, hyper-representations do not collapse to a single solution, but decode to diverse and useful weights.

### 4.2.3 Sampling Initializations for Transfer Learning

**Setup:** We investigate the effectiveness of our method in a transfer-learning setup across image datasets. In particular, we report transfer learning results from SVHN to MNIST and from STL-10 to CIFAR-10 as two representative scenarios. Results on the other pairs of datasets can be found in Appendix F. In these experiments, pre-trained models $B_F$ and the hyper-representation model are trained on a source domain. Subsequently, the pre-trained models $B_F$ and the samples $S_{\text{KDE}}$, $S_{\text{Neigh}}$ and $S_{\text{GAN}}$ are fine-tuned on the target domain. The baseline approach ($B_T$) is based on training models from scratch on the target domain.

Table 2: Transfer-learning results (mean and standard deviation of the test accuracy in %). Note that for STL-10 to CIFAR-10 the performance of all methods saturate quickly due to the limited capacity of models in the zoo making further improvements challenging as we discuss in § 4.3.

| Method | SVHN to MNIST | | | STL-10 to CIFAR-10 | | |
|---|---|---|---|---|---|---|
| | Ep. 0 | Ep. 1 | Ep. 50 | Ep. 0 | Ep. 1 | Ep. 50 |
| $B_T$ | $10.0 \pm 0.6$ | $20.6 \pm 1.6$ | $91.1 \pm 1.0$ | $10.1 \pm 1.3$ | $27.5 \pm 2.1$ | $48.7 \pm 1.4$ |
| $B_F$ | $\mathbf{33.4 \pm 5.4}$ | $84.4 \pm 7.4$ | $95.0 \pm 0.8$ | $\mathbf{15.3 \pm 2.3}$ | $29.4 \pm 1.9$ | $\mathbf{49.2 \pm 0.7}$ |
| $S_{\text{KDE30}}$ | $31.8 \pm 5.6$ | $\mathbf{86.9 \pm 1.4}$ | $\mathbf{95.5 \pm 0.4}$ | $14.5 \pm 1.9$ | $\mathbf{29.6 \pm 2.0}$ | $48.8 \pm 0.9$ |
| $S_{\text{Neigh30}}$ | $10.7 \pm 2.7$ | $79.2 \pm 3.3$ | $\mathbf{95.5 \pm 0.7}$ | $10.1 \pm 2.1$ | $29.2 \pm 1.9$ | $48.9 \pm 0.7$ |
| $S_{\text{GAN30}}$ | $10.4 \pm 2.4$ | $75.0 \pm 6.3$ | $94.9 \pm 0.7$ | $10.2 \pm 2.5$ | $28.6 \pm 1.8$ | $48.8 \pm 0.8$ |

**Results:** When transfer learning is performed from SVHN to MNIST, the sampled populations on average learn faster and achieve significantly higher performance than the $B_T$ baseline and generally compares favorably to $B_F$ (Figure 7, Table 2). In the STL-10 to CIFAR-10 experiment, all populations appear to saturate with only small differences in their performances (Table 2). Different sampling methods perform differently at the beginning versus the end of transfer learning. Generally, $S_{\text{KDE30}}$ performs better in the first epochs, while all methods perform comparably at the end of transfer-learning. These discrepancies underline the difficulty of developing a single strong sampling method, which is an interesting area of future research. We further found that all datasets are useful sources for all targets (see Appendix F). Interestingly and other than in related work [29], even transfer from the simpler to harder datasets (e.g., MNIST to SVHN) improves performance. This might be explained by the ability of hyper-representations to capture a generic inductive prior useful across different domains, which we further investigate next.

**Conditioning on unseen zoos:** We explore if the hyper-representation trained on the models of one zoo (e.g. MNIST) can reconstruct the weights of another unseen zoo (e.g. SVHN). This can be useful to enable generation of weights for novel tasks without the need to retrain a hyper-representation. This is analogous to instance-conditioned GANs that recently were able to generate images from unseen domains without retraining GANs [2]. Our results in Table 3 show that while the performance on the unseen zoos is reduced, it is still well above random guessing (10%), especially when multiple model weights are sampled and ensembled. This is promising, as the hyper-representations were trained on single-dataset zoos.

Table 3: Test accuracy (%) of models generated conditioned on the models of unseen zoos.

| Training zoo | Conditioning (unseen) | Mean / max (bolded) accuracy | |
|---|---|---|---|
| | | One model | Ensemble |
| MNIST | SVHN | 12.7 / **19.8** | 13.4 / **18.7** |
| SVHN | MNIST | 16.2 / **26.0** | 22.1 / **29.8** |
| CIFAR-10 | STL-10 | 18.0 / **24.4** | 23.8 / **26.7** |
| STL-10 | CIFAR-10 | 16.3 / **21.2** | 20.0 / **23.0** |

### 4.2.4 Sampling Initializations for Unseen Architectures

Generalization to unseen large architectures with complex connectivity (ResNet, MobileNet, and EfficientNet) is a very interesting and ambitious research problem. As a step towards that goal, we perform experiments in which we attempted to use our hyper-representation beyond the same simple architecture. Surprisingly, our results indicate the promise of leveraging the hyper-representation for more diverse architectures and settings. Further experiments investigating the cross-architecture generalization capabilities of hyper-representations can be found in Appendix D.

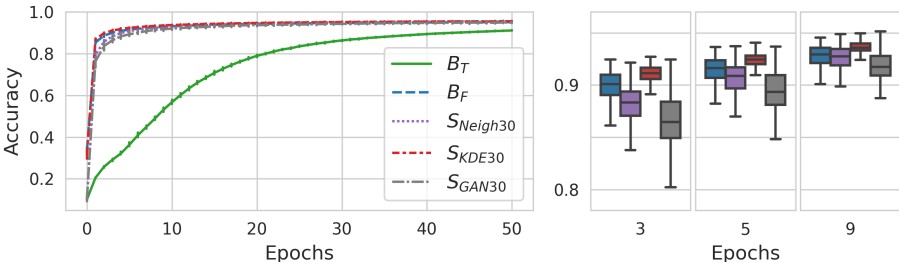

Figure 7: SVHN to MNIST transfer learning experiment: test accuracy over epochs. Our sampling methods outperform the baselines after the first epoch. **Left:** epochs from 0 to 50. **Right**: epochs from 3 to 9, where $B_T$ is significantly lower than 80% and thus is not visible.

**Setup:** With this experiment, we aim to verify if it is possible to adapt our approach to architectures not seen during training, e.g., with skip connections and/or with more layers. We follow the transfer-learning setup of § 4.2.3 and use an existing MNIST hyper-representation to sample weights as initializiation for training on SVHN. However, we now also vary the architecture. While the decoder outputs a fixed-sized vector of weights, we can assign these weights to new architectures by either making sure that the new architecture still has the same number of parameters or by initializing randomly the extra parameters introduced. Specifically, we create three cases: (1) we add ResNet-style skip connections [16] (1x1 conv) to the convolutional layers (3-conv + res-skip), (2) re-distribute the weights to smaller four convolutional layers (4-conv), (3) re-distribute to smaller four convolutional layers and add identity skip connections (4-conv + id.-skip).

**Results:** Surprisingly, despite training our hyper-representation on the models of the same architecture, generated weights for all three cases outperform random initialization and converge significantly faster across all the variations (Table 4). In all the variations even just after 5 epochs the models with generated weights are better than training the baseline for 50 epochs. In the 3-conv + res-skip experiments, some models in both populations did not learn, which leads to high standard deviation. Further analysis is required to explain the

Table 4: Test accuracy (%) on SVHN of populations with generated weights compared to models trained from scratch $B_T$. Best results for each epoch and dataset are bolded. `r. i.` indicates random initialization, `gen.` denotes weights generated with our ($S_{KDE30}$).

| Initialization | Epoch 1 | Epoch 5 | Epoch 50 |
|---|---|---|---|
| 3-conv (r. i.) + res-skip (r. i.) | 18.9 ± 1.6 | 31.4 ± 17 | 50.6 ± 28 |
| 3-conv (gen.) + res-skip (r. i.) | **34.5 ± 14** | **60.5 ± 21** | **68.0 ± 21** |
| 4-conv (r. i.) | 19.2 ± 1.0 | 19.2 ± 0.9 | 55.2 ± 11 |
| 4-conv (gen.) | **44.0 ± 4.5** | **57.8 ± 3.5** | **67.6 ± 1.9** |
| 4-conv + id.-skip (r. i.) | 18.9 ± 1.0 | 19.6 ± 1.7 | 56.4 ± 7.9 |
| 4-conv + id.-skip (gen.) | **48.0 ± 4.0** | **59.9 ± 2.5** | **66.4 ± 1.7** |

gains of our approach in this challenging setup. To extend and scale up our method further, future work could combined it with the methods of growing networks [3, 42], so that some layers are generated while some are initialized in a sophisticated way to preserve the functional form of the network.

## 4.3 Limitations of Zoos with Small Models

To thoroughly investigate different methods and make experiments feasible, we chose to use the model zoos of the same small scale as in [37]. While on MNIST and SVHN, the architectures of such model zoos allowed us to achieve high performance, on CIFAR-10 and STL-10, the performance of all populations is limited by the low capacity of the models zoo's architecture. The models saturate at around 50% and 40% accuracy, respectively. The sampled populations reach the saturation point and fluctuate, but cannot outperform the baselines, see Appendix F for details. We hypothesize that due to the high remaining loss, the weight updates are correspondingly large without converging or improving performance. This may cause the weights to contain relatively little signal and high noise. Larger model architectures might mitigate this behaviour. Corresponding model zoos have recently been made available in [38] to tackle this issue[1].

---

[1] www.modelzoos.cc

# 5 Related Work

**HyperNetworks:** Recently, representation learning on neural networks is typically based on HyperNetworks that learn low-dimensional structure of model weights to generate weights in a deterministic fashion [14, 1, 21, 45]. HyperNetworks have also been extended to meta-learning by conditioning weight generation on data [46, 36]. Closely related to our work, HyperGANs [35] can sample model weights by combining the hypernetworks and the GAN framework. Similarly, [8] allow for sampling model weights by conditioning the hypernetwork on a noise vector. However, training hypernetwork-based methods require input data (e.g. images) to feed to the neural networks. In practice, there may already be large collections of trained models, while their training data may not always be accessible. Learning representations of model weights without data, called hyper-representations, has been recently introduced in [37]. Our methods build on that work to allow for better reconstruction and sampling. [7] showed that given a few parameters of a network, the remaining values of a single model can be accurately reconstructed. However, in our work we leverage the autoencoder to train a representation of the entire model zoo. Very recently, [33] use diffusion on a population of models to generate model weights for the original task via prompting.

**Transfer Learning:** Transfer learning via fine-tuning aims at re-using models and their learned knowledge from a source to a target task [44, 4, 9, 29, 22]. Transfer learning models makes training less expensive, boosts performance, or allows to train on datasets with very few samples and has been applied on a wide range of domains [48]. The common transfer learning methods however only consider transferring from a single model, and so disregard the large variety of pre-trained models and potential benefit of combining them.

**Knowledge distillation:** Our work is related to [43, 25, 39] that allow to distill knowledge from a model zoo into a single network. Knowledge distillation overcomes the inherent limitation of transfer learning by transferring the knowledge from many large teacher models to a relatively small student model [25, 39]. Knowledge distillation however requires the source models at training as in [25] and at inference as in[39] thus increasing memory cost. Further, the learned knowledge cannot be shared between different target models. **Learnable initialization** [6, 47] provide methods to improve initialization by leveraging the meta-learning and gradient-flow ideas. In contrast to knowledge distillation and learnable initialization, we train a hyper-representation of a model zoo in a latent space, which is a more general and powerful approach that can enable sampling an ensemble, property estimation, improved initialization and implicit knowledge distillation across datasets.

# 6 Conclusion

In this paper, we propose a new method to sample from hyper-representations to generate neural network weights in one forward pass. We extend the training objective of hyper-representations by a novel layer-wise loss normalization which is key to the capability of generating functional models. Our method allows us to generate diverse populations of model weights, which show high performance as ensembles. We evaluate sampled models both in-dataset as well as in transfer learning and find them capable to outperform both models trained from scratch, as well as pre-trained and fine-tuned models. Populations of sampled models, even for some unseen architectures, generally learn faster and achieve statistically significantly higher performance. This demonstrates that such hyper-representation can be used as a generative model for neural network weights and therefore might serve as a building block for transfer learning from different domains, meta learning or continual learning.

## Acknowledgments

This work was partially funded by Google Research Scholar Award, the University of St.Gallen Basic Research Fund, and project PID2020-117142GB-I00 funded by MCIN/ AEI /10.13039/501100011033. We are thankful to Michael Mommert for editorial support.

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
