# A  Model Zoo Details

The model zoos are generated following the method of [37, 38] An overview of the model zoos is given in in Table 5. All model zoos share one general CNN architecture, outlined in Table 6. The hyperparameter choices for each of the population are listed in Table 7. The hyperparameters are chosen to generate zoos with smooth, continuous development and spread in performance.

Table 5: Model zoo overview.

| Zoo | Input Channels | Parameters | Population Size |
|---|---|---|---|
| MNIST | 1 | 2464 | 1000 |
| SVHN | 1 | 2464 | 1000 |
| CIFAR-10 | 3 | 2864 | 1000 |
| STL-10 | 3 | 2864 | 1000 |

Table 6: CNN architecture details for the models in model zoos.

| Layer | Component | Value |
|---|---|---|
| Conv 1 | input channels | 1/3 |
| | output channels | 8 |
| | kernel size | 5 |
| | stride | 1 |
| | padding | 0 |
| Max Pooling | kernel size | 2 |
| Activation | tanh / gelu | |
| Conv 2 | input channels | 8 |
| | output channels | 6 |
| | kernel size | 5 |
| | stride | 1 |
| | padding | 0 |
| Max Pooling | kernel size | 2 |
| Activation | tanh / gelu | |
| Conv 3 | input channels | 6 |
| | output channels | 4 |
| | kernel size | 2 |
| | stride | 1 |
| | padding | 0 |
| Activation | tanh / gelu | |
| Linear 1 | input channels | 36 |
| | output channels | 20 |
| Activation | tanh / gelu | |
| Linear 2 | input channels | 20 |
| | output channels | 10 |

Table 7: Hyperparameter choices for the model zoos.

| Model Zoo | Hyperparameter | Value |
|---|---|---|
| MNIST | input channels | 1 |
| | activation | tanh |
| | weight decay | 0 |
| | learning rate | 3e-4 |
| | initialization | uniform |
| | optimizer | Adam |
| | seed | [1-1000] |
| SVHN | input channels | 1 |
| | activation | tanh |
| | weight decay | 0 |
| | learning rate | 3e-3 |
| | initialization | uniform |
| | optimizer | adam |
| | seed | [1-1000] |
| CIFAR-10 | input channels | 3 |
| | activation | gelu |
| | weight decay | 1e-2 |
| | learning rate | 1e-4 |
| | initialization | kaiming-uniform |
| | optimizer | adam |
| | seed | [1-1000] |
| STL-10 | input channels | 3 |
| | activation | tanh |
| | weight decay | 1e-3 |
| | learning rate | 1e-4 |
| | initialization a | kaiming-uniform |
| | optimizer | adam |
| | seed | [1-1000] |

# B  Hyper-Representation Architecture and Training Details

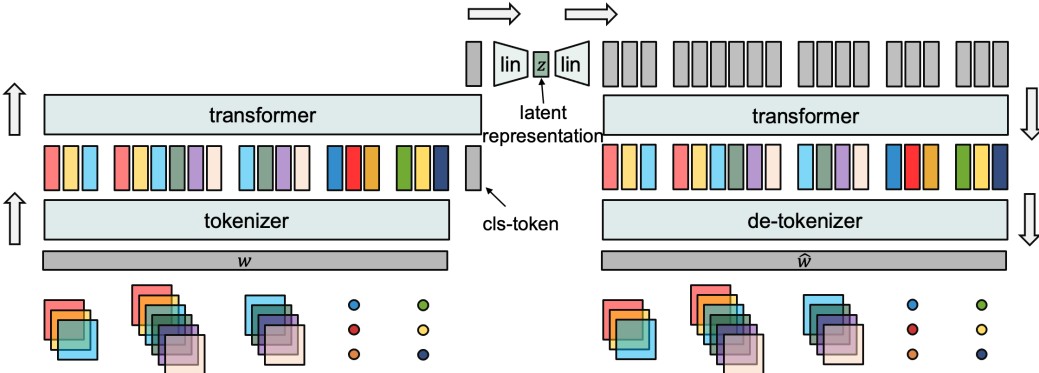

Figure 8: Schematic of the auto-encoder architecture to learn hyper-representations.

Hyper-representations are learned with an autoencoder based on multi-head self-attention. The architecture is outlined in Figure 8. Convolutional and fully connected neurons are embedded to token embeddings of dimension $d_{token}$. Learned position encodings are added to provide relational information. A learned compression token (CLS) is appended to the sequence of token embeddings. The sequence of token embeddings is passed to $N_{layers}$ layers of multi-head self-attention with $N_{heads}$ heads with hidden embedding dimension $d_{hidden}$. The CLS token is compressed to the bottleneck of dimension $d_z$ with an MLP or a linear layer. For the decoder, an MLP or a linear layer maps the bottleneck to a sequence of token embeddings. The sequence is passed through another stack of multi-head self-attention, which is symmetric to the encoder. Debedders map the token embeddings back to convolutional and fully connected neurons. The reconstruction and contrastive loss are balanced with a parameter $\beta$. The contrastive loss is computed on the embeddings $\mathbf{z}$ mapped through a projection head $\bar{\mathbf{z}} = p(\mathbf{z})$, where $p$ is a learned MLP with four layers with 400 neurons each and $\bar{\mathbf{z}}$ has 50 dimensions. In Table 8, the exact hyper-parameters for each of the hyper-representation are listed to reproduce our results.

Table 8: Hyper-representation architecture and training details.

|  | MNIST | SVHN | CIFAR-10 | STL-10 |
|---|---|---|---|---|
|  | *Architecture* | | | |
| $d_{inpot}$ | 2464 | 2464 | 2864 | 2864 |
| $d_{token}$ | 972 | 1680 | 1488 | 1632 |
| $d_{hidden}$ | 1140 | 1800 | 1164 | 1680 |
| $N_{layers}$ | 2 | 4 | 2 | 4 |
| $N_{heads}$ | 12 | 12 | 12 | 24 |
| $d_z$ | 700 | 1000 | 700 | 700 |
| Compression | linear | linear | linear | linear |
|  | *Training* | | | |
| Optimizer | Adam | Adam | Adam | Adam |
| Learning rate | 0.0001 | 0.0001 | 0.0001 | 0.0001 |
| Dropout | 0.1 | 0.1 | 0.1 | 0.1 |
| Weight Decay | 1e-09 | 1e-09 | 1e-09 | 1e-09 |
| $\beta$ | 0.977 | 0.920 | 0.950 | 0.950 |
| training epochs | 1750 | 1750 | 500 | 2000 |
| batch size | 500 | 250 | 200 | 200 |

# C   Evaluation of Layer-Wise Loss Normalization

To evaluate layer-wise loss normalization, we compare two hyper-representations with comparable reconstruction. Both have a $R^2 = 1 - \frac{mse(\hat{\mathbf{w}}, \mathbf{w})}{mse(\mathbf{w_{mean}}, \mathbf{w})}$ as a measure of the explained variance of around 70%. One is trained trained with the baseline hyper-representation MSE, the other with layer-wise-normalization. Figures 9 and 10 show the distribution of weights per layer before and

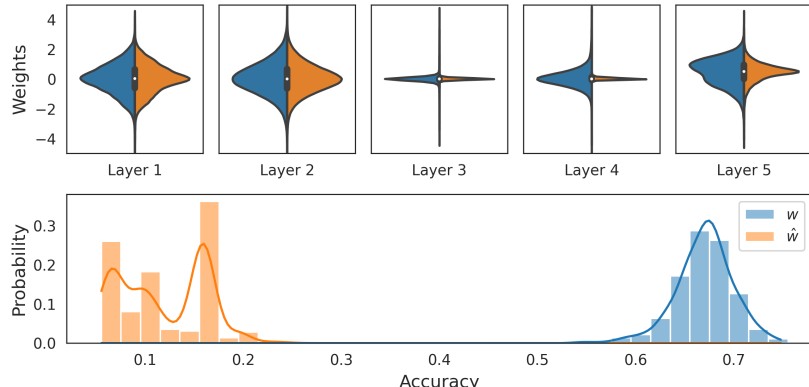

Figure 9:   **Top:** Weight distribution per layer (1-5) of the SVHN test set before $w$ and after reconstruction $\hat{w}$ with the basline hyper-representation training loss. Layers 3 and 4 have small weight distributions, therefore add little penalty to the MSE and are consequently poorly reconstructed. **Bottom:** Accuracy distribution of the same population before and after reconstruction. The badly reconstructed layers (top) cause the reconstructed models to perform around random guessing.

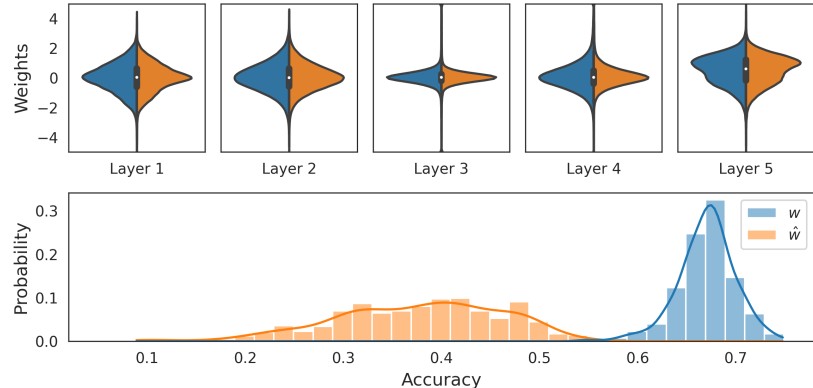

Figure 10:   **Top:** Weight distribution per layer (1-5) of the SVHN test set before $w$ and after reconstruction $\hat{w}$ with layer-wise loss normalization. The distributions of all layers are more similar, the reconstruction is equally distributed across the layers. **Bottom:** Accuracy distribution of the same population before and after reconstruction. The normalization fixes the catastrophic failure of the models. The remaining loss in accuracy can be explained with remaining reconstruction error.

after reconstruction, as well as the accuracy distribution of both populations on the SVHN image test set. With the basline learning scheme in Figure 9, the distributions in layers 3 and 4 do not match. In these layers, the original weight distribution is smaller, and so there is only a small error even if the reconstructions predicts the mean. These layers become a weak link of the reconstructed models, and cause performance around random guessing. With layer-wise loss normalization in Figure 10, the weight distribution between the layers becomes more similar. As a consequence, the reconstruction error is more evenly distributed across the layers, there are no single layers that aren't reconstructed at all. This appears to allow information to flow forward through the model, and significantly improves the performance of reconstructed models. We find layer-wise-normalization necessary to reconstruct or sample functional models across all populations, where the weights are unevenly distributed.

# D  Hyper-Representation Analysis

In this section, we detail the analysis of hyper-representations. We begin with their geometry, followed by the distributions of individual dimensions of hyper-representations, and finally investigate robustness and smoothness.

**Embeddings in Hyper-Representation Space Populate a Hyper-Sphere**  We analyse the geometry of hyper-representations $\mathbf{z}$. The space of hyper-representations is bounded to a high dimensional box by a tanh activation. Surprisingly, hyper-representations do not populate the entire space, but sections on a shell of a high-dimensional sphere. Figure 11 shows the distribution of the norm of the embeddings of the MNIST zoo. All embeddings are distributed on a small band between length 10 and 12, therefore they must populate the shell of a hyper-sphere. In Figure 12 we investigate pairwise cosine distances between the embeddings of the MNIST zoo. The majority of the embeddings populate the region between 0.6 and 0.8. The outliers around 1.0 are embeddings of the same model at different epochs. This indicates that models are not entirely orthogonal, but mutually equally far apart, populating a section of the shell of the hyper-sphere. While hyper-spheres are commonly found in embeddings of contrastive learning [19], in our experiments hyper-spheres form even without a contrastive loss. Properties of the models embedded on that hyper-sphere can be predicted from hyper-representations, therefore the topology on the sphere appears to encode model properties.

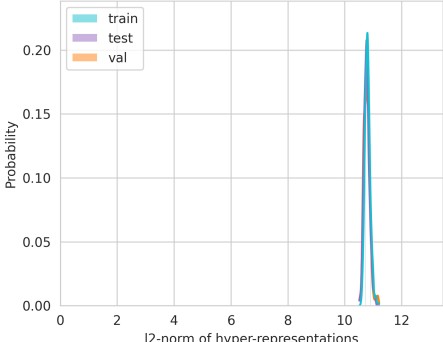 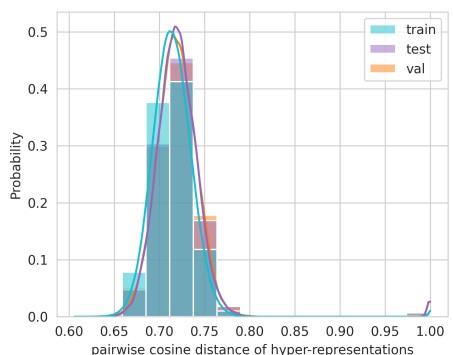

Figure 11: Distributions of $\ell_2$ norm of hyper-representations $\mathbf{z}$ of the MNIST zoo.

Figure 12: Distributions of pairwise cosine distance of hyper-representations $\mathbf{z}$ of the MNIST zoo.

**Distributions of Dimensions of Embeddings in Hyper-Representation Encode Properties**  Previous work showed that linear probing from hyper-representations accurately predicts i.e. model accuracy. In these linear probes, the individual $z$ dimensions each linearly contribute to accuracy predictions. This allows us investigate $z$ dimensions independently. Figure 13 shows examples for the distribution of selected individual dimensions of hyper-representations $\mathbf{z}$. On the left are the distribution of the entire population, on the right of the top 30 % performing models. The individual dimensions show different types of distributions, with different modes. Most have a zero mean and span 3/4 of the available range, but some collapse to either $-1$ or $1$. Further, the distributions also differ in at least some dimension between the entire population, and the better performing split of the population.

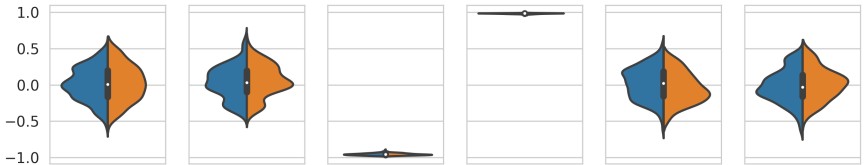

Figure 13: Distributions of individual dimensions of hyper-representations $\mathbf{z}$ of the MNIST zoo. In blue is the distribution of all samples, in orange the subset of the 30 % best samples.

**Generalization Capabilities of Hyper-Representations to Diverse Model Zoos** There are certain architectural changes such as adding/removing/changing pooling layers and nonlinearity that do not change the number of parameters (the dimensionality of the input/output required by our approach). These changes as well as changes of hyperparameters used to train models in a zoo may drastically alter the distribution of weights and pose a challenge to the proposed approach. Modern neural networks (ResNet, MobileNet, EfficientNet, etc.) are often trained with very different hyperparameters. With the experiment below, we investigate the generalization capabilities of hyper-representations to suchchanges, which might be important for modern large-scale settings as well.

**Setup:** We experimentally evaluate generalizability of the proposed approach on models trained with a different choice of nonlinearity or other hyperparameters with two experiments (a and b). To that end, in addition to the original SVHN test zoo (zoo 1), we use two more diverse SVHN zoos (zoo 2 and zoo 3). In zoo 2, in addition to random seed, models differ in the activation (tanh, relu, gelu, sigmoid), l2-regularization (0, 0.001, 0.1) and dropout (0,0.3,0.5). In zoo 3 (extending zoo 2), we increase the diversity further by additionally varying the initialization method (uniform, normal, kaiming-uniform, kaiming-normal) and the learning rate (0.0001, 0.001, 0.01).

**Experiment (a):** We first evaluate our original encoder-decoder trained on a model zoo varying in random seed only. For evaluation, we pass the test splits of zoo 2 and zoo 3 through the encoder-decoder. We measure the reconstruction $R^2$ score of the original encoder-decoder on the diverse test zoos.
**Results:** Our results (Table D) indicate that our original encoder-decoder can still encode and decode weights even in such a challenging setting, although there is an expected drop of performance.

**Experiment (a):** We next evaluate if hyper-representations can be trained on diverse zoos. For this experiment, we train a hyper-representation on the train split of zoo 3. With this, we aim to show that training hyper-representations on diverse zoos improves generalization capabilities further.
**Results:** Our results show that training on diverse zoos is a much more difficult task to optimize, hence the reconstruction on the original zoo degrades. It nonetheless improves the reconstruction results on the test split of the diverse zoos 2 and 3. This indicates that varying seeds and hyperparameters may be different aspects of complexity that need to be considered.

Table 9: Generalizability of hyper-representations towards more diverse model zoo configurations (measured as the reconstruction score, higher is better).

| Training zoo | Test zoo 1: original | Test zoo 2: vary activation | Test zoo 3: vary hyperparameters |
|---|---|---|---|
| Original | 81.9% | 45.7% | 38.9% |
| Diverse (zoo 3) | 25.8% | 89.1% | 75.6% |

# E  Sampling Methods

## E.1  VAE

A common extension of the autoencoder of [37] to enable sampling from its latent representation is to make the autoencoder variational [20]. In our experiments, VAEs could not be trained to satisfactory reconstruct model weights without unweighting the KL-divergence to insignificance essentially making it deterministic as in [37]. Empirically, embeddings in hyper-representations are mapped on the shell of a sphere (see Section D) and leave the inside of the sphere entirely empty. On the other hand, a gaussian prior allocates most of the probability mass near the center of the sphere. It therefore appears plausible that the two may be incompatible. That issue of non-compatible priors is well known. [10] find that regularizing embeddings and decoder yields equally smooth representation spaces as VAEs without restrictions to specific priors. During training of hyper-representations, both encoder and decoder are regularized with a small $\ell_2$ penalty. Further, dropout is applied throughout the autoencoder, which servers as another regularizer and adds blurryness to the embeddings. The combination of dropout, the erasing augmentation and the contrastive loss further regularizes the hyper-representation space. In all our sampling methods, we draw samples from probability distributions, which effectively disconnects the drawn samples from training embeddings.

## E.2  Latent Space GAN Details

The generator and discriminator of our GAN consist of four fully-connected layers interleaved with ReLU nonlinearities. The same architecture and training hyperparameters are used for all experiments. The generator's input is a Gaussian noise $\mathbf{n}^*$ of dimensionality $d = 16$, the hidden dimensionalities are 128, 256 and 512, and the output dimensionality is equal to the hyper-representation length $D$. The discriminator's input is $D$-dimensional, the hidden dimensionalities are 1024, 512 and 256, and the output dimensionality is a scalar denoting either a real or fake sample. The discriminator is regularized with Spectral Norm [31]. The discriminator and generator are trained for 1000 epochs and batch size 32 using Adam with a two time-scale update rule [18]: learning rate is 1e-4 for the generator and 2e-4 for the discriminator.

# F Full Experiment Results

## F.1 Digit Domain

Table 10: Accuracy of sampled models: median and 95% confidence intervals. On the main diagonal are in-dataset experiments, otherwise transfer-learning from source to target. Bold numbers highlight the best source-to-target results. N/A enotes cases, in which the boot-strapped CI on the median could not be computed.

| Population | Source | Target | |
|---|---|---|---|
| | | MNIST | SVHN |
| $B_T$ | | 91.1 [91.1, 91.2] | 72.3 [72.0, 72.4] |
| $B_F$ | | 91.2 [91.0, 91.3] | 76.2 [75.8, 76.5] |
| $S_{\text{KDE}}$ | | 92.3 [92.1, 92.8] | 76.7 [76.2, 77.0] |
| $S_{\text{KDE30}}$ | MNIST | 93.1 [92.9, 93.4] | 77.2 [76.8, 77.6] |
| $S_{\text{Neigh}}$ | | 93.4 [93.2, 93.5] | 76.8 [76.4, 77.1] |
| $S_{\text{Neigh30}}$ | | **94.0 [93.8, 94.1]** | **77.0 [76.3, 77.4]** |
| $S_{\text{GAN}}$ | | 93.5 [93.3, 93.6] | 76.9 [76.6, 77.6] |
| $S_{\text{GAN30}}$ | | 93.9 [93.5, 93.9] | 76.5 [76.3, 76.8] |
| $B_F$ | | 95.1 [95.0, 95.3] | 73.2 [72.8, 73.4] |
| $S_{\text{KDE}}$ | | 95.1 N/A | 73.0 [72.6, 73.3] |
| $S_{\text{KDE30}}$ | | 95.5 N/A | 74.2 [73.9, 74.5] |
| $S_{\text{Neigh}}$ | SVHN | **97.2 [97.0, 97.3]** | **78.1 [77.9, 78.2]** |
| $S_{\text{Neigh30}}$ | | 95.5 [95.4, 95.7] | 76.5 [76.3, 76.7] |
| $S_{\text{GAN}}$ | | 94.3 [94.1, 94.6] | 74.5 [74.0, 74.9] |
| $S_{\text{GAN30}}$ | | 94.9 [94.8, 95.1] | 75.3 [75.0, 75.6 |

Table 11: Mann-Whitney U test of Samples S vs Baselines B: p-value and CLES (Common Language Effect Size). p-values indicate the probability of the samples of two groups originating from the same distribution. CLES=0.5 indicates no effect, CLES=1.0 a strong positive, CLES=0.0 a strong negative effect. As the results indicate, both proposed sampling methods are almost always statistically significantly better than the two baselines. Further, their effect is often very strong.

| Population Pairs | Source | Target | |
|---|---|---|---|
| | | MNIST | SVHN |
| $S_{\text{KDE}}$ vs. $B_T$ | | **2.1e-18 \| 0.8701** | **5.2e-27 \| 0.9551** |
| $S_{\text{KDE}}$ vs. $B_F$ | | **0.0e+00 \| 0.8639** | **1.1e-01 \| 0.5920** |
| $S_{\text{KDE30}}$ vs. $B_T$ | | **7.0e-27 \| 0.9539** | **2.5e-29 \| 0.9754** |
| $S_{\text{KDE30}}$ vs. $B_F$ | | **6.9e-22 \| 0.9545** | **1.7e-04 \| 0.7180** |
| $S_{\text{Neigh}}$ vs. $B_T$ | | **1.5e-30 \| 0.9857** | **6.6e-31 \| 0.9888** |
| $S_{\text{Neigh}}$ vs. $B_F$ | MNIST | **4.5e-25 \| 0.9889** | **5.2e-03 \| 0.6622** |
| $S_{\text{Neigh30}}$ vs. $B_T$ | | **1.7e-35 \| 0.9987** | **1.3e-29 \| 0.9778** |
| $S_{\text{Neigh30}}$ vs. $B_F$ | | **3.1e-28 \| 0.9994** | **1.4e-02 \| 0.6426** |
| $S_{\text{GAN}}$ vs. $B_T$ | | **7.6e-31 \| 0.9883** | **8.0e-25 \| 0.9351** |
| $S_{\text{GAN}}$ vs. $B_F$ | | **3.0e-25 \| 0.9907** | **7.8e-03 \| 0.6546** |
| $S_{\text{GAN30}}$ vs. $B_T$ | | **1.1e-31 \| 0.9953** | **2.1e-26 \| 0.9496** |
| $S_{\text{GAN30}}$ vs. $B_F$ | | **6.8e-26 \| 0.9973** | **4.9e-02 \| 0.6144** |
| $S_{\text{KDE}}$ vs. $B_T$ | | **6.1e-79 \| 0.9943** | **1.1e-04 \| 0.6006** |
| $S_{\text{KDE}}$ vs. $B_F$ | | 7.8e-01 \| 0.4904 | 3.8e-01 \| 0.4704 |
| $S_{\text{KDE30}}$ vs. $B_T$ | | **1.7e-82 \| 1.0000** | **1.6e-30 \| 0.7985** |
| $S_{\text{KDE30}}$ vs. $B_F$ | | **0.0e+00 \| 0.7292** | **3.0e-08 \| 0.6850** |
| $S_{\text{Neigh}}$ vs. $B_T$ | | **2.9e-78 \| 0.9867** | **8.6e-80 \| 0.9916** |
| $S_{\text{Neigh}}$ vs. $B_F$ | SVHN | **2.8e-44 \| 0.9661** | **1.8e-47 \| 0.9833** |
| $S_{\text{Neigh30}}$ vs. $B_T$ | | **1.7e-82 \| 1.0000** | **4.7e-76 \| 0.9797** |
| $S_{\text{Neigh30}}$ vs. $B_F$ | | **8.2e-08 \| 0.6791** | **1.7e-42 \| 0.9563** |
| $S_{\text{GAN}}$ vs. $B_T$ | | **1.2e-31 \| 0.9948** | **0.0e+00 \| 0.8140** |
| $S_{\text{GAN}}$ vs. $B_F$ | | 1.5e-07 \| 0.2517 | **7.5e-06 \| 0.7118** |
| $S_{\text{GAN30}}$ vs. $B_T$ | | **4.2e-32 \| 0.9987** | **6.7e-22 \| 0.9067** |
| $S_{\text{GAN30}}$ vs. $B_F$ | | 3.6e-01 \| 0.4565 | **0.0e+00 \| 0.8335** |

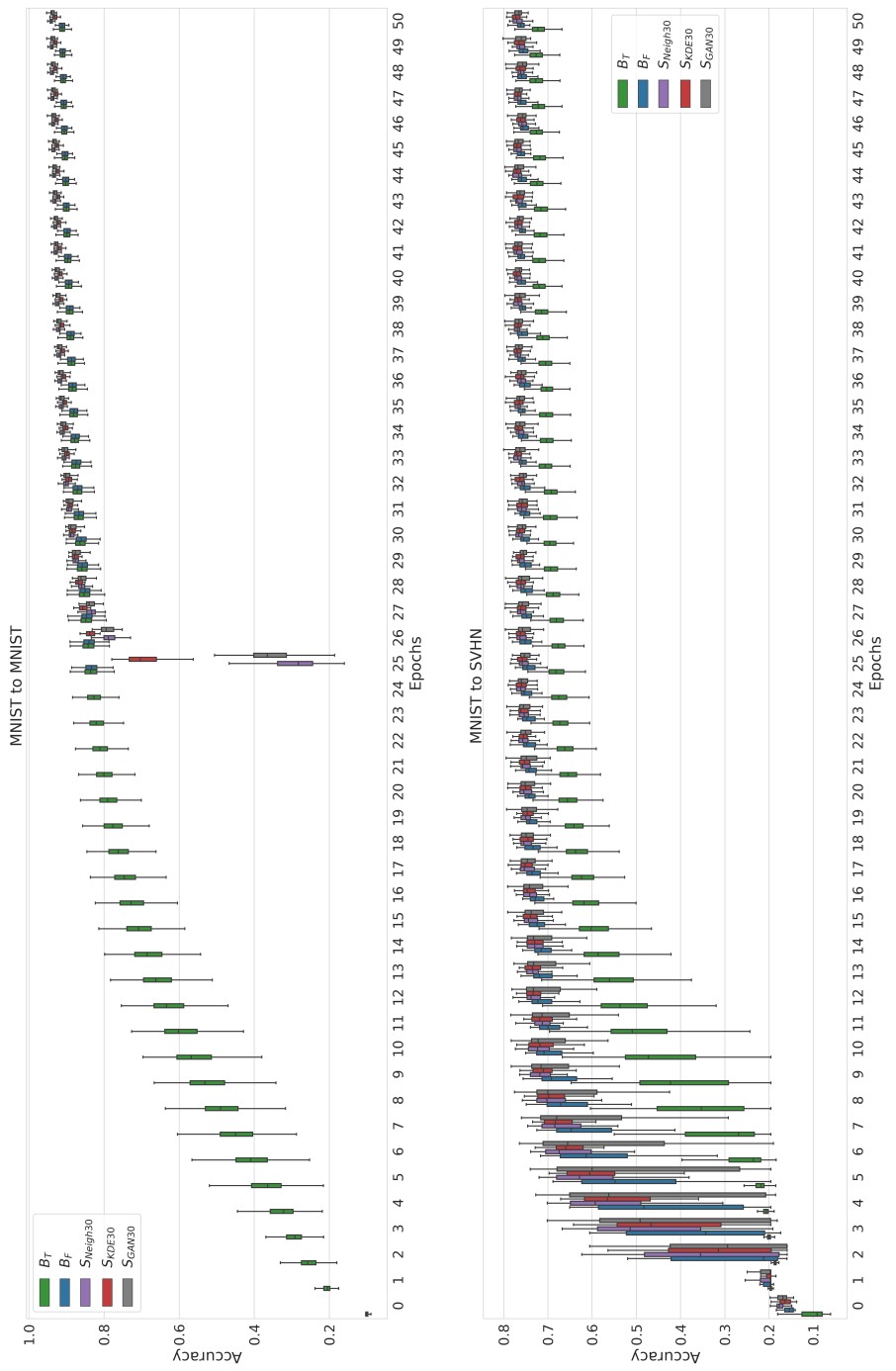

Figure 14: MNIST in-dataset experiment: accuracy over epochs. Boxes indicate quintiles 25 to 75.

Figure 15: MNIST to SVHN transfer learning experiment: accuracy over epochs. Boxes indicate quintiles 25 to 75.

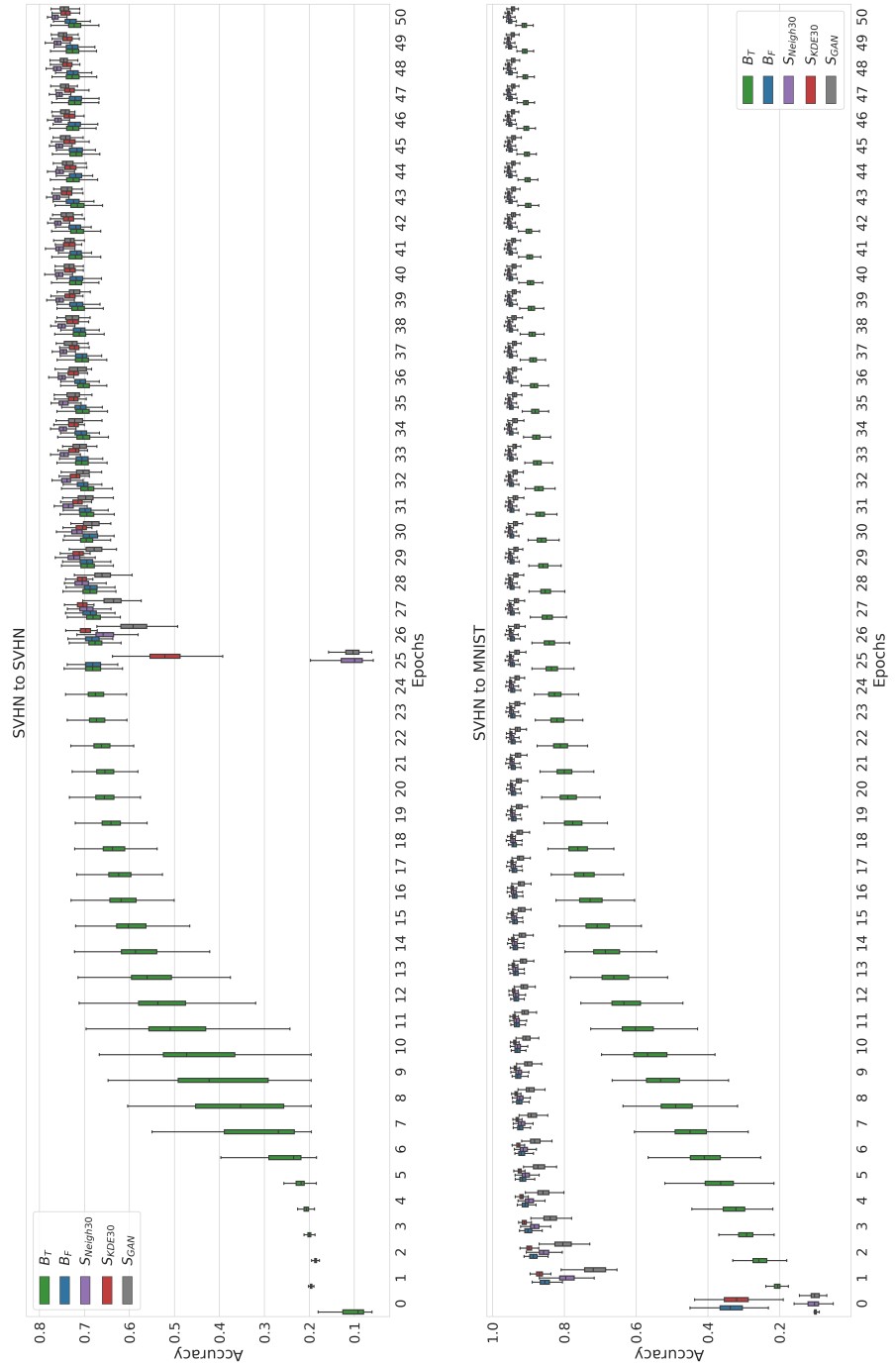

Figure 16: SVHN in-dataset experiment: accuracy over epochs. Boxes indicate quintiles 25 to 75.

Figure 17: SVHN to MNIST transfer learning experiment: accuracy over epochs. Boxes indicate quintiles 25 to 75.

## F.2 Natural Images Domain

Table 12: Accuracy of sampled models: median and 95% confidence intervals. On the main diagonal are in-dataset experiments, otherwise transfer-learning from source to target. Bold numbers highlight the best source-to-target results. N/A enotes cases, in which the boot-strapped CI on the median could not be computed.

| Population | Source | Target | |
| --- | --- | --- | --- |
| | | CIFAR-10 | STL-10 |
| $B_T$ | | 49.0 [48.9, 49.0] | 39.0 [38.9, 39.1] |
| $B_F$ | | 48.6 [48.3, 48.7] | **42.8 [42.5, 42.9]** |
| $S_{KDE}$ | | 48.3 [48.1, 48.4] | 40.7 [40.3, 40.9] |
| $S_{KDE30}$ | CIFAR-10 | **48.7 [48.4, 48.8]** | 41.3 [40.9, 41.5] |
| $S_{Neigh}$ | | 45.6 [44.9, 46.0] | 36.7 [35.8, 37.4] |
| $S_{Neigh30}$ | | 46.2 [45.8, 46.4] | 37.9 [37.3, 38.2] |
| $S_{GAN}$ | | 46.0 N/A | 38.6 [38.1, 39.0] |
| $S_{GAN30}$ | | 47.0 [46.5, 47.2] | 38.6 [38.2, 39.1] |
| $B_F$ | | **49.3 [49.0, 49.4]** | **39.5 [38.9, 39.7]** |
| $S_{KDE}$ | | 48.6 [48.4, 48.9] | 37.3 [37.0, 37.8] |
| $S_{KDE30}$ | | 48.8 [48.4, 49.2] | 38.3 [37.9, 38.4] |
| $S_{Neigh}$ | STL-10 | 10.0 N/A | 28.3 [26.8, 29.1] |
| $S_{Neigh30}$ | | 49.0 [48.5, 49.1] | 37.8 [37.6, 38.2] |
| $S_{GAN}$ | | 49.0 [48.6, 49.4] | 38.5 [37.9, 38.9] |
| $S_{GAN30}$ | | 48.8 [48.5, 49.1] | 37.9 N/A |

Table 13: Mann-Whitney U test of Samples S vs Baselines B: p-value and CLES (Common Language Effect Size). p-values indicate the probability of the samples of two groups originating from the same distribution. CLES=0.5 indicates no effect, CLES=1.0 a strong positive, CLES=0.0 a strong negative effect.

| Population Pairs | Source | Target | |
| --- | --- | --- | --- |
| | | CIFAR-10 | STL-10 |
| $S_{KDE}$ vs. $B_T$ | | 1.5e-06 \| 0.2966 | **7.4e-19 \| 0.8750** |
| $S_{KDE}$ vs. $B_F$ | | 3.7e-02 \| 0.4014 | 1.7e-18 \| 0.0849 |
| $S_{KDE30}$ vs. $B_T$ | | 3.6e-02 \| 0.4114 | **4.8e-25 \| 0.9371** |
| $S_{KDE30}$ vs. $B_F$ | | **2.9e-01 \| 0.5498** | 0.0e+00 \| 0.1266 |
| $S_{Neigh}$ vs. $B_T$ | | 5.7e-28 \| 0.0364 | 7.4e-18 \| 0.1359 |
| $S_{Neigh}$ vs. $B_F$ | CIFAR-10 | 3.1e-22 \| 0.0413 | 7.1e-26 \| 0.0024 |
| $S_{Neigh30}$ vs. $B_T$ | | 3.5e-25 \| 0.0616 | 2.0e-07 \| 0.2800 |
| $S_{Neigh30}$ vs. $B_F$ | | 2.2e-19 \| 0.0741 | 3.0e-25 \| 0.0089 |
| $S_{GAN}$ vs. $B_T$ | | 6.6e-25 \| 0.0642 | 6.6e-02 \| 0.4223 |
| $S_{GAN}$ vs. $B_F$ | | 2.8e-19 \| 0.0754 | 1.0e-24 \| 0.0145 |
| $S_{GAN30}$ vs. $B_T$ | | 2.1e-21 \| 0.0983 | 1.1e-02 \| 0.3928 |
| $S_{GAN30}$ vs. $B_F$ | | 8.8e-16 \| 0.1195 | 2.7e-25 \| 0.0084 |
| $S_{KDE}$ vs. $B_T$ | | 1.3e-01 \| 0.4362 | 0.0e+00 \| 0.1730 |
| $S_{KDE}$ vs. $B_F$ | | 6.9e-04 \| 0.3028 | 6.0e-10 \| 0.1404 |
| $S_{KDE30}$ vs. $B_T$ | | 6.1e-01 \| 0.4783 | 1.2e-06 \| 0.2948 |
| $S_{KDE30}$ vs. $B_F$ | | 1.1e-02 \| 0.3528 | 9.1e-06 \| 0.2424 |
| $S_{Neigh}$ vs. $B_T$ | | 2.9e-32 \| 0.0000 | 3.0e-32 \| 0.0000 |
| $S_{Neigh}$ vs. $B_F$ | | 3.3e-20 \| 0.0000 | 7.1e-18 \| 0.0000 |
| $S_{Neigh30}$ vs. $B_T$ | STL-10 | 1.0e+00 \| 0.5000 | 4.3e-09 \| 0.2517 |
| $S_{Neigh30}$ vs. $B_F$ | | 2.1e-02 \| 0.3654 | 5.4e-07 \| 0.2090 |
| $S_{GAN}$ vs. $B_T$ | | 3.2e-01 \| 0.5418 | 2.0e-04 \| 0.3427 |
| $S_{GAN}$ vs. $B_F$ | | 2.7e-01 \| 0.4360 | 2.4e-04 \| 0.2864 |
| $S_{GAN30}$ vs. $B_T$ | | 6.2e-01 \| 0.4788 | 5.4e-07 \| 0.2880 |
| $S_{GAN30}$ vs. $B_F$ | | 1.2e-02 \| 0.3532 | 4.6e-06 \| 0.2340 |

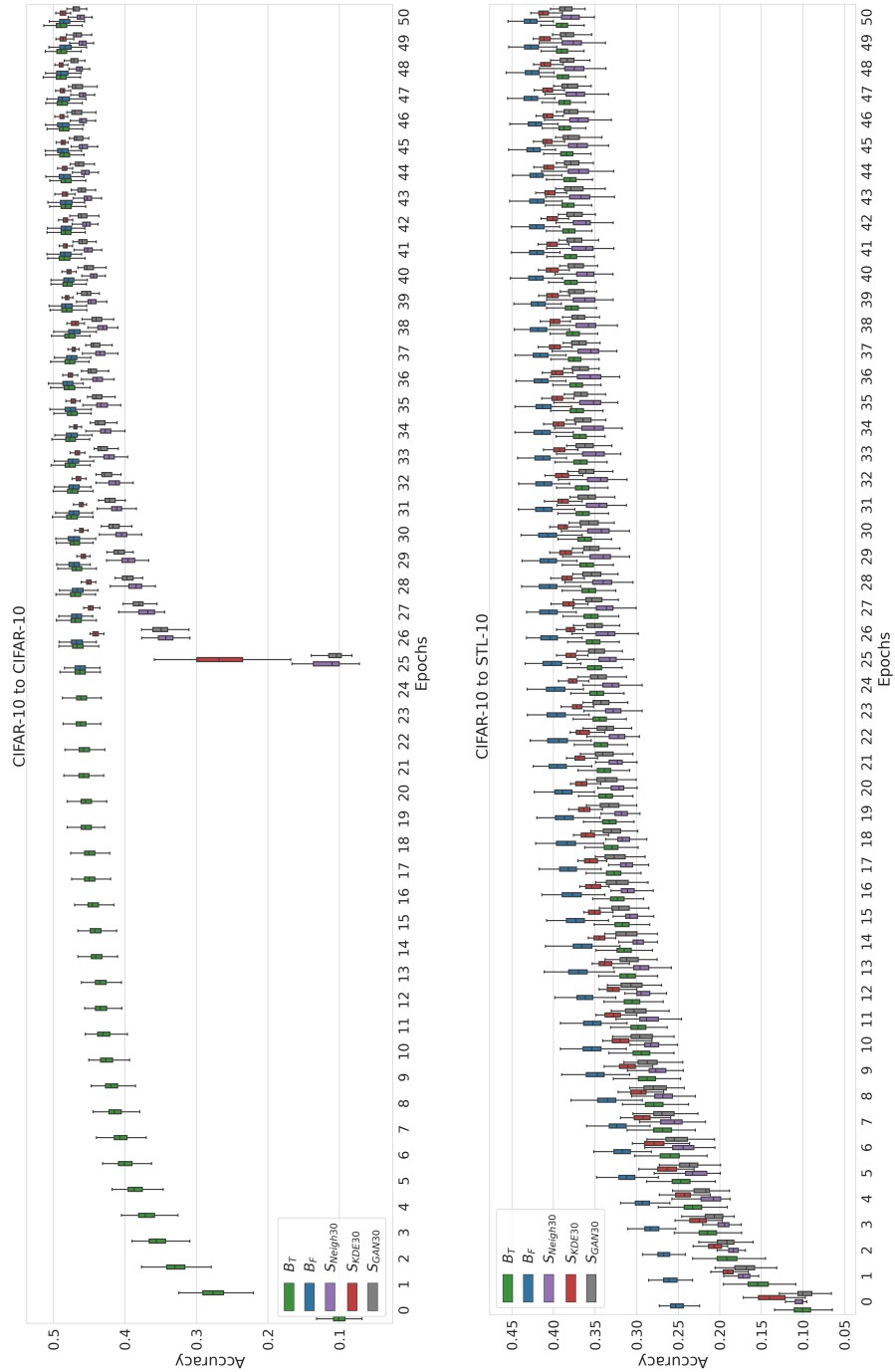

Figure 18: CIFAR-10 in-dataset experiment: accuracy over epochs. Boxes indicate quintiles 25 to 75.

Figure 19: CIFAR-10 to STL-10 transfer learning experiment: accuracy over epochs. Boxes indicate quintiles 25 to 75.

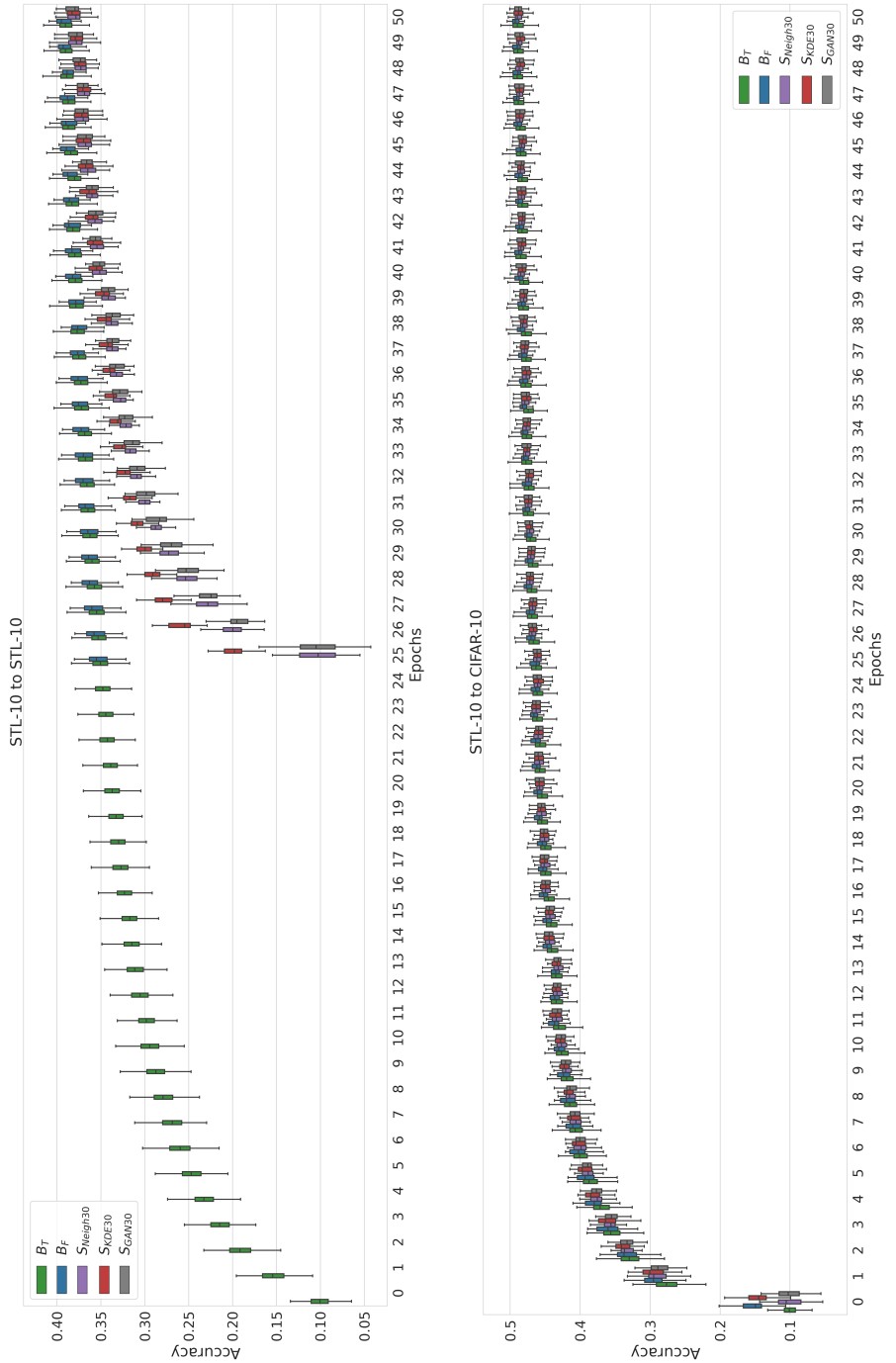

Figure 20: STL-10 in-dataset experiment: accuracy over epochs. Boxes indicate quintiles 25 to 75.

Figure 21: STL-10 to CIFAR-10 transfer learning experiment: accuracy over epochs. Boxes indicate quintiles 25 to 75.