# OpenReview forum: "Hyper-Representations as Generative Models: Sampling Unseen Neural Network Weights"
_NeurIPS.cc/2022/Conference — NeurIPS 2022 Accept_

### Official Review · Reviewer_cFnj · 2022-07-11

**Rating:** 6
**Confidence:** 3
**Soundness:** 4 excellent
**Presentation:** 3 good
**Contribution:** 3 good

**Summary:**

The core idea of this paper is using a collection of pre-trained neural networks (here classifier "zoos") to learn a generative model capturing the structure of the "weight manifold" of these networks. This generative model can later be used to generate good weight initializations, or model ensembles. Several practical techniques for capturing these weight distributions are discussed and compared to each other (as well as to some natural baselines).


**Questions:**

1. The proposed approach allows the generative model to capture information about numerous pre-trained classifiers, essentially learning some information about the "weight manifold" of networks with reasonable performance. Yet in empirical comparisons (see, for example, Section 4.2.3), baseline models are individual pre-trained networks ($B_F$) that seemingly lack any information about the local structure in the weight space. An alternative baseline that could capture some information about the weight manifold, could, for example, be based on an ensemble of pre-trained models that are later fine-tuned on a new task and finally averaged (either in the weight space, which would require similar or identical initialization and would be more fair, or even averaged predictions like in a conventional ensemble). Fine-tuning a collection of models is, of course, more expensive, but does not require an initial investment in building a model zoo. I am curious if the authors agree that it would be an interesting baseline that could perhaps be studied later?
2. If I am not mistaken, there is an interesting empirical observation that the loss function is typically smooth and well-behaved on a linear segment (in the weight space) connecting two different models trained with identical, or just similar initializations (also for model checkpoints at different training steps). This fact could have implications for some observations in Section 4.2.1 (the authors observed that the loss changes smoothly along linear segments in the weight embedding space). I am curious if the authors think that these two observations could be connected and could point at approximate global linearity of the "embedding to weight" map?
3. It appears that contrastive loss would typically favor the situation where equivalent network weights (up to a channel permutation) are mapped to the same embedding. This in turns means that corresponding generative model would typically be unable to generate different equivalent weight configurations. I wonder if authors find this to be a limiting or advantageous property for generating ensembles of models?


**Limitations:**

The proposed technique shows promising results, but it's current practical applications could be limited by a very significant computational investment necessary to train a model zoo.

**Strengths And Weaknesses:**

### Strengths

1. The overall idea of learning the structure of the model weight manifold and training generative models to sample from this manifold appears very interesting and intriguing, at least conceptually.
2. The paper outlines an effective layer-wise loss normalization technique and several different approaches for learning such generative models (using various limiting simplifications, but overall very practical). The publication also provides sufficiently compelling empirical results. It is inspiring to see that trained generative models can capture the structure of the weight manifold and learn the set of sufficiently accurate neural networks, even capable of producing competitive model ensembles.

### Weaknesses

1. The proposed technique could be an invaluable tool for understanding the topology and properties of the weight manifold and it even shows some practical promise, but in its current form it appears to require a very significant computational investment for pre-training an ensemble zoo. This large computational cost could make it somewhat impractical for real-life applications.
2. Section 3.2.3 proposes to use "an invertible function" from $\mathbb{R}^D$ to $\mathbb{R}^d$. I do not think that for $D>d$ such smooth invertible functions exist. In my opinion, the explanation in Section 3.2.3 could be clarified and made more rigorous. It is not entirely clear to me how $d$-dimensional samples $n_i$ are mapped back to the $D$-dimensional space.

---

> ### Author Response · Authors · 2022-08-02
> **Response to Reviewer cFnj**
>
> We thank the reviewer for the valuable review and insightful questions. We will answer them individually below.
>
> > W1: "It appears to require a very significant computational investment for pre-training an ensemble zoo"
>
> We kindly refer to **Shared concern of afuw and cFnj: “high cost of collecting a model zoo”** in the **Response to all Reviewers - Part 2**. for a discussion about the need of a model zoo and associated computational costs.
>
> > W2: "The explanation in Section 3.2.3 could be clarified and made more rigorous"
>
> Thank you for raising this point. Indeed, instead of “invertible” we mean “approximate inverse”. We will correct that in the manuscript.
> The method is motivated by the observation that embeddings populate the hyper-representation space very sparsely. Naive sampling will therefore very often sample low-likelihood regions, as the counterfactual results show in Figure 4 of the submission. As the representation space encodes model properties, the neighborhood relation of embeddings seems meaningful. We therefore use a neighborhood-based dimensionality reduction to map the embeddings to a lower dimensional space, which has fewer low-probability regions and thus simplifies sampling. To that end, we use UMAP, which is designed to preserve local and global neighborhood and offers inverse mappings, see e.g. [UMAP]. The inverse is only approximate, and does introduce noise particularly at high compression ratios. However, empirically the noise introduced by neighborhood-based sampling appears not harmful, as S_Neigh30 performs very competitively to other sampling methods (see our results in the main text and Appendix).
>
> [UMAP] "How UMAP Works" at umap-learn.readthedocs.io
>
> ### Question 1 - Ensemble baselines
>
> Adding baselines with more context in the weight space is a very interesting idea. The ensembling experiments (Figure 5 in the submission) show that ensembling the logits improves the performance for all populations that are diverse and have non-trivial performance.
> From our perspective, mixing the weights of different models may be difficult. Due to the inherent symmetries, such a mixture would most likely converge to a sample from the weight’s distribution, which we expect to hurt the performance. Constraining the models to the same seed, similar to self-ensembling [Terv17], may be a good way to solve that problem, which we leave for future work.
>
> [Terv17] Mean teachers are better role models: Weight-averaged consistency targets improve semi-supervised deep learning results. Tarvainen, Valpola, NeurIPS 2017.
>
>
> ### Question 2 - Global linearity
>
> There is research investigating the paths in weight space, e.g. [Lucas21], which indeed finds smoothness and monotonicity in loss along linear interpolations between initialization and last epoch, similar to our finding in embedding space (Figure 3 in the submission). Whether that indicates a global notion of approximate linearity of the mapping is a difficult and interesting question. Intuitively, given the inherent nonlinearity of neural networks, we assume structures in weight space to be nonlinear, too. Similarly, our architecture mapping weights to embeddings is nonlinear. However, the high dimensionality may indeed render the problem approximately linear, in an extension of [Dauph14]. We will keep this in mind for future work.
>
> [Lucas21] On Monotonic Linear Interpolation of Neural Network Parameters, Lucas et al., ICML 2021
>
> [Dauph14] Identifying and attacking the saddle point problem in high-dimensional non-convex optimization, Dauphin et al., NeurIPS 2014
>
> ### Question 3 - Equivalent network weights
>
> To us, the different symmetric points in weight space that relate to equivalent networks can be seen as different instanciations of the same function or mapping represented by the neural network model. Similar to rotated (natural) images, they have different data representations, but the same semantic concept. As the permuted versions of networks represent the same function with the same errors, ensembling them unfortunately brings no benefits.
> In our view, these equivalent networks increase the complexity of the problem. Bronstein et al. have presented a formalized approach to such equivariances and invariances [Bronst21] that may be of use to formalize the problem of learning representations of neural network weights. The contrastive loss is another way to deal with that complexity and reduce its impact on the latent representation. It can be seen as a weak proxy to a canonical representation. That way, the contrastive loss is designed to increase the functional diversity of samples, as it becomes unlikely to sample instances of the same function.
>
> [Bronst21] Geometric Deep Learning: Grids, Groups, Graphs, Geodesics, and Gauges, Bronstein et al., 2021
>
> We hope to have addressed the raised points and answered the questions. We’d gladly hear of any other questions, and look forward to the discussion.

---

> > ### Comment · Reviewer_cFnj · 2022-08-08
> > **Reply to the authors**
> >
> > I would like to thank the authors for the detailed reply that clarified most of my questions and concerns!

---

> > > ### Author Response · Authors · 2022-08-08
> > > **Response #2 to Reviewer cFnj**
> > >
> > > We thank the reviewer for the response, we are happy to have clarified most of the questions. We are happy to address anything that remains unclear or answer follow-up questions and look forward to the rest of the discussion.

---

### Official Review · Reviewer_nctY · 2022-07-11

**Rating:** 7
**Confidence:** 3
**Soundness:** 3 good
**Presentation:** 4 excellent
**Contribution:** 3 good

**Summary:**

This paper proposes different methods to sample model weights from a VAE trained on network parameters given by a model zoo. To improve sample quality and smoothness of the hyper-representation space, the reconstruction loss is scaled differently for each layer of the architecture (coined layer-wise loss normalization). Experiments confirm that the hyper-representation space is smoother with this loss normalization. In a wide range of experiments, the paper shows that by initializing network parameters with this VAE networks are trained faster with improved performance over training from scratch.

**Questions:**

Typos:
- Line 56: "a self attention blocks"
- Line 57: "Each of the these"

**Limitations:**

The paper discusses the main limitation of the paper ({W1}): The limited capacity of the models in the zoo.

**Strengths And Weaknesses:**

Strengths:
- {S1} The paper is well-structured, well-written, and easy to follow.
- {S2} The approach tackles the significant problem of learning representations of network parameters and shows that initializing parameters with the learned model indeed is advantageous in terms of training time and final network performance.
- {S3} The paper shows lots of experiments that investigate the applicability of the learned hyper-representation space.

Weaknesses:
- {W1} The main weakness of the paper is that the learned hyper-representation space is only usable for the same single architecture that the model zoo is comprised of. This substantially reduces the practicability of the method and reduces the challenge of learning the space.

---

> ### Author Response · Authors · 2022-08-02
> **Response to Reviewer nctY**
>
> We thank the reviewer for the valuable review.
>
> > W1: "The main weakness of the paper is that the learned hyper-representation space is only usable for the same single architecture that the model zoo is comprised of. This substantially reduces the practicability of the method and reduces the challenge of learning the space."
>
> The limitation to a fixed architecture is aimed at controlling the experiment design and facilitating robust and reproducible development, which may be scaled up in future work. Nevertheless, we were able to extend our approach to more diverse architectures and settings as we show in the **Response to all Reviewers** with **Experiment 1** and **Experiment 2**.
>
> We hope our response could address the raised point, we welcome any new questions and look forward to the discussion.

---

### Official Review · Reviewer_afuw · 2022-07-12

**Rating:** 4
**Confidence:** 4
**Soundness:** 2 fair
**Presentation:** 3 good
**Contribution:** 2 fair

**Summary:**

This work proposes to encode the network weights of a set of models into “hyper-representations”, which can be decoded to generate network weights (i.e., new models). The authors show that weights generated by the proposed hyper-representations are better than random initialization and baseline hyper-representations.

This work proposes layer-wise loss normalization to learn better hyper-representations and several sampling methods to decode weights from the learned hyper-representation.

Experimental results on CIFAR, MNIST, SVHN, and STL-10 are provided to demonstrate the usefulness of the proposed method on three tasks: (1) network weights initialization, (2) model ensembles, and (3) transfer learning.


**Questions:**

See above for details.


**Limitations:**

Yes

**Strengths And Weaknesses:**

Encoding a set of trained neural networks into compact representations and applying such representations to other tasks is an interesting topic. Networks trained from large-scale datasets clearly contain meaningful knowledge of the data. Training such representations is a possible way for us to extract the knowledge from trained networks and transfer the knowledge to other tasks. From this perspective, this work is of interest to the community.

However, I have concerns about the usefulness of the proposed method and the rigorousness of the empirical evaluation.

1. This work only considers a simple architecture - 3 Conv + 2 Fully-Connected Layers. It’s unclear whether the proposed idea is useful on modern network architectures with complex skip connections or branches, e.g., ResNet, MobileNet, and EfficientNet.

2. Also, all networks in the model zoo are restricted to having the same architectures. I feel this is a big limitation. What if we want to generate the weights of a different architecture? It seems there are no trivial solutions that can fix this.

3. The hyper-representations need to be trained on a set of trained models (e.g., 1000), which can be expensive if the dataset is large or the network is large. This work only has results on small datasets. For image classification, results on ImageNet are highly preferred. While I understand that collecting 1000 networks on ImageNet is indeed expensive, this implies the intrinsic limitation of the proposed idea.

4. For model ensembles, we usually directly train multiple models with different random seeds and compute the ensemble. This is much cheaper and easier than collecting a large set of trained networks. Same for transfer learning, we can pre-train the model on dataset A and then finetune on dataset B. Why do we need to use such hyper-representations?

Overall, I am not convinced by the usefulness of the proposed idea. Therefore, I recommend rejection.

---

> ### Author Response · Authors · 2022-08-02
> **Response to Reviewer afuw**
>
> We thank the reviewer for the valuable review. We appreciate that the topic is interesting and is of interest to the research community.
>
> > W1: “It’s unclear whether the proposed idea is useful on modern network architectures with complex skip connections or branches, .e.g, ResNet, MobileNet, and EfficientNet.”
>
> We were able to extend our approach to simple ResNet-style architectures by adding skip-connections, please see **Experiment 1** (Table 1) and the discussion therein in the **Response to all Reviewers - Part 1**.
> Scaling up our approach to million parameter networks (e.g., ResNet, MobileNet, and EfficientNet) is not the focus of this work, but we are looking forward to future works in that direction.
>
> > W2: "The zoos only contain models of the same architecture, there are no trivial solutions that can fix this."
>
> We address the concern in **Experiment 1** and **Experiment 2** (Tables 1 and 2 respectively) and the discussion therein in the **Response to all Reviewers**.
>
>
> > W3: "The hyper-representations need to be trained on a set of trained models (e.g., 1000)"
>
> Please see **Shared concern of afuw and cFnj: “high cost of collecting a model zoo”** in  **Response to all Reviewers - Part 2** for a discussion about the need of a model zoo and associated computational costs.
>
>
> > W4: "Why do we need to use such hyper-representations?"
>
> Generally, learning representations of neural network weights is one way of studying emergent structures, of which understanding seems both interesting and useful. More immediately practical and much more efficient is the scenario of re-using hyper-representations for new datasets. As our experiments in Sec. 4.2.3 show, the proposed method can generate models for new datasets without the need to train on them. We showed the results indicating such a possibility in Table 3 in the submission. Beyond that, conceptually, hyper-representations allow to combine knowledge from many models, potentially trained on different domains or datasets.
>
> We hope we could clarify the raised issues. If there are any new questions, we are happy to answer them, and look forward to the discussion!

---

> > ### Comment · Reviewer_afuw · 2022-08-07
> > **Comments after reading the rebuttal**
> >
> > Thanks for providing the detailed response and extensive results. I really appreciate the hard work the authors put in the rebuttal. The Experiment 1 & 2 added in the rebuttal are very informative. I have upgraded my score from rejection to borderline reject.
> >
> > The following concerns prevent me from giving a definite acceptance.
> >
> > All the experiments still are conducted on very simple architectures (even after considering the new results in the rebuttal). It's understandable to claim that "Scaling up our approach to million parameter networks (e.g., ResNet, MobileNet, and EfficientNet) is not the focus of this work", but those million parameter networks are what people actually use in practice.
> >
> > All the results are obtained on small datasets. Most model zoos available online, e.g., https://pytorch.org/serve/model_zoo.html, focus on large-scale datasets like ImageNet. The conclusion will be much more solid if we can see evidence on larger datasets.
> >
> > > Generally, learning representations of neural network weights is one way of studying emergent structures, of which understanding seems both interesting and useful.
> >
> > Conceptually, I could accept this statement. But the meaning of "emergent structures" is vague. Is that convolution or what it could be? How does this relate to neural architecture search? There could be a remote possibility that such hyper-representations can be useful to create new architectures. I understand that these are all open problems and it's totally unfair to require one work to answer them all. But using that to support this work is not convincing.
> >
> > Overall, I think this work is a good exploration of learning hyper-representations of network weights. But I am not convinced by the usefulness or significance of the method or results.

---

> > > ### Author Response · Authors · 2022-08-08
> > > **Response #2 to Reviewer afuw**
> > >
> > > We thank reviewer “afuw” for the feedback, we highly appreciate the response and score upgrade as well as the acknowledgement of the additional experiments. We would like to clarify the remaining three points.
> > >
> > >
> > > >1. “but those million parameter networks are what people actually use in practice.”
> > >
> > > We understand the concerns reviewer “afuw” has regarding the model complexity and size and would like to respond in two parts.
> > > In several research communities small-scale datasets and small models are actively used and are defining the state-of-the-art in these communities, e.g., in meta and continual learning, e.g., [Finn17, Zhmo22, Rame22]. Here, relatively simple datasets like Omniglot, or CIFAR-100 or down-scaled mini-imagenet are still used. In the AutoML / Neural Architecture Search community, the [NASBench dataset family](https://www.automl.org/nas-overview/nasbench/) is very popular. These datasets contain performance metrics of comparably small CNN models trained on datasets like CIFAR-10 or Fashion MNIST, e.g. [YING19]. Other fundamental research on symmetric structures of neural networks rely on small MLP models [Sims21]. In these communities, small models and relatively simple tasks are being actively used to develop and evaluate methods, to iterate quickly and at low computational cost.
> > > As already mentioned in our initial response above, we are not aware of any methods in the structured generative models area that provide sufficient scalability for our purposes. Our problem size is of the same order of magnitude as the upper end of what the graph/set representation learning community is achieving [Dai20]. Scaling up to the complexity of, e.g., ResNets exceeds the current state of the art by several orders of magnitude. Such feats are in our opinion too ambitious for a single submission like ours. We believe our work can make a challenging and well-motivated use case for structured generative models and we hope we can excite more researchers of this novel topic to enable faster progress.
> > >
> > > [Finn17] Model-Agnostic Meta-Learning for Fast Adaptation of Deep Networks. Finn et al., ICML 2017
> > > [YING19] NAS-Bench-101: Towards Reproducible Neural Architecture Search. Ying et al., PMLR 2019
> > > [Zhmo22] HyperTransformer: Model Generation for Supervised and Semi-Supervised Few-Shot Learning. Zhmoginov et al., ICML 2022.
> > > [Rame22] Model Zoo: A Growing "Brain" That Learns Continually. Ramesh and Chaudhari, ICLR 2022.
> > >
> > >
> > > >2. “All the results are obtained on small datasets”
> > >
> > > As we state in our response above, these small datasets are still actively used in several research communities. Once the feasibility of scaling graph/set representation learning to higher dimensional problems is demonstrated, we hope future work by us or others can address zoos trained on large datasets. We have discussed related limitations in Limitations (Sec 4.3 in the submission) and will update its content given these concerns.
> > >
> > >
> > > >3. “But the meaning of "emergent structures" is vague. Is that convolution or what it could be? How does this relate to neural architecture search?“
> > >
> > > Re-reading our answer above, we acknowledge that our take on “emergent structures” may not be as accurate as we intended it to be. By that we refer to the structures in the **distribution** of weights given a population of trained models. These structures appear to exist and encode properties of the models ([36] [25] [33] in our submission) or define subspaces [Sims21, Wort21, Luca21], which we find fascinating. Identifying such structures, learning how they relate to model properties (e.g. accuracy or generalization gap) and how to instantiate them for new models (for transfer learning or ensemble generation) seems both relevant and significant to us. We evaluate and discuss some use cases in the submission. Generating useful weights for unseen architectures in NAS is another very interesting use-case, we show early results of applications to varying architectures in our Response to all Reviewers above that may be extended to that end.
> > >
> > > [Sims21] Geometry of the Loss Landscape in Overparameterized Neural Networks: Symmetries and Invariances. Simsek et al., ICML 2021.
> > > [Wort21] Learning Neural Network Subspaces. Wortsman et al., ICML 2021.
> > > [Luca21] On Monotonic Linear Interpolation of Neural Network Parameters. Lucas et al., ICML 2021.
> > >
> > >
> > > We hope to have addressed the concerns and look forward to the remaining discussion!

---

### Author Response · Authors · 2022-08-02
**Response to all Reviewers - Part 1**

We thank all the three reviewers (**afuw**, **nctY**, **cFnj**) for their valuable feedback. We are glad that all the three reviewers highlighted the strengths of our paper: “an interesting topic” (**afuw**), “significant problem” (**nctY**), “very interesting and intriguing idea” (**cFnj**); “lots of experiments” (**nctY**), “overall very practical” (**cFnj**) and “compelling empirical results” (**cFnj**).

Below we first address **two common concerns** of the reviewers. We split the common response into two parts due to space constraints. We then respond to the reviewers’ specific concerns under the corresponding reviews. We hope our response addresses the questions of the reviewers and improves the submission and we look forward to the discussion.

## 1. Shared concern by afuw and nctY: “applicability/usefulness for modern and diverse architectures”

Generalization to unseen large architectures with complex connectivity (ResNet, MobileNet, and EfficientNet) is a very interesting and ambitious research problem to address next. As a step towards that goal, we performed **Experiment 1** and **Experiment 2** in which we attempted to use our hyper-representation beyond the same simple architecture. Surprisingly, our results indicate the promise of leveraging the hyper-representation for more diverse architectures and settings (see details below).

### Experiment 1: Extension of Sec 4.2.3 (Sampling for Cross Dataset Initialization) to Unseen Architectures

**Setup.** In the first experiment, we aim to verify if it is possible to adapt our approach to architectures not seen during training, e.g., with skip connections and/or with more layers. We follow the setup of Sec 4.2.3 in our submission and use an existing MNIST hyper-representation to sample weights as initializiation for training on SVHN. Now we also vary the architecture. The idea is that even though our decoder always outputs a flat 2464-dimensional (on this task) vector of weights, we can assign these weights to more than “3 Conv + 2 Fully-Connected Layers” by either making sure that the new architecture still has the same number of parameters or by initializing randomly the extra parameters introduced. We explore these variations and present the results in **Table 1** below.

- In the first variation, we extend our original architecture with additional ResNet-style skip-connections (**3-conv + skip**). The skip-connections contain a 1x1 convolution layer (as in ResNets) to match channel dimensionalities, which we initialize randomly.
- In the second variation, we add a fourth convolution layer (**4-conv**), but keep the overall number of parameters equal by reducing the dimensionality in convolutional and fully-connected layers. This approach is naive, as that the encoder is unaware of the changes in the architecture, so for example some weights that are intended for conv layer 3 will be used in conv layer 4.
- The third variation extends the second one by adding identity (without 1x1 conv) skip-connections to the 4-conv setup (**4-conv + identity skip**), also preserving the overall number of parameters.

In all variations, we compare with random init.

**Table 1. Initializing networks with skip connections (classification accuracies on SVHN).**
| Initialization                                         | Epoch 1       | Epoch 5       | Epoch 50      |
|--------------------------------------------------------|---------------|---------------|---------------|
| 3-conv (random init) + skip (random init)  | 18.9 ± 1.6 %  | 31.4 ± 17.9 % | 50.6 ± 27.6%  |
| 3-conv (generated) + skip (random init) | **34.5** ± 14.4 % | **60.5** ± 21.3 % | **68.0** ± 21.2%  |
| 4-conv (random init)                                   | 19.2 ± 1.0 %  | 19.2 ± 0.9 %  | 55.2 ± 11.0 % |
| 4-conv (generated)                                     | **44.0** ± 4.5 %  | **57.8** ± 3.5 %  | **67.6** ± 1.9 %  |
| 4-conv + identity skip (random Init)       | 18.9 ± 1.0 %  | 19.6 ± 1.7 %  | 56.4 ± 7.9 %  |
| 4-conv + identity skip (generated)         | **48.0** ± 4.0 %  | **59.9** ± 2.5 %  | **66.4** ± 1.7 %  |

**Results.** Surprisingly, despite the limitations of our approach, generated weights outperform random init and converge significantly faster across all the variations. In all the variations even just after 5 epochs the models with generated weights are better than training the baseline for 50 epochs. Further analysis is required to explain the gains of our approach in this challenging setup.
To extend and scale up our method further, one can combine it with the methods of growing networks [Chen16, Wang21], so that some layers are generated while some are initialized in a sophisticated way to preserve the functional form of the network.

[Chen16] Net2Net: Accelerating Learning via Knowledge Transfer,Chen,Goodfellow,Shlens, ICLR 2016

[Wang21] Recurrent Parameter Generators, Wang,Chen,Yu,Cheung,LeCun, arXiv:2107.07110

See our **Response to all Reviewers - Part 2** next.

---

> ### Author Response · Authors · 2022-08-02
> **Response to all Reviewers - Part 2**
>
> ### Experiment 2: Generalizing Hyper-Representations to Diverse Model Zoo Configurations
>
> There are certain architectural changes such as adding/removing/changing pooling layers and nonlinearity that do not change the number of parameters (the dimensionality of the input/output required by our approach). These changes as well as changes of hyperparameters used to train models in a zoo may drastically alter the distribution of weights and pose a challenge to the proposed approach. Modern neural networks (ResNet, MobileNet, EfficientNet, etc.) are often trained with very different hyperparameters, so the second experiment outlined below might be important for modern large-scale settings as well.
>
> **Setup.** We experimentally evaluate generalizability of the proposed approach on models trained with a different choice of nonlinearity or other hyperparameters with two experiments (2a and 2b). To that end, in addition to the original test zoo (zoo 1), we use two more diverse zoos (zoo 2 and zoo 3). In zoo 2, in addition to random seed, models differ in the activation (tanh, relu, gelu, sigmoid), l2-regularization (0, 0.001, 0.1) and dropout (0,0.3,0.5). In zoo 3 (extending zoo 2), we increase the diversity further by additionally varying the initialization method (uniform, normal, kaiming-uniform, kaiming-normal) and the learning rate (0.0001, 0.001, 0.01).
>
>
> (**2a**). We first evaluate our original encoder-decoder trained on a model zoo varying in random seed only. For evaluation, we pass the test splits of zoo 2 and zoo 3 through the encoder-decoder. We measure the reconstruction R^2 score of the original encoder-decoder on the diverse test zoos.
> **Results.** Our results (Table 2) indicate that our original encoder-decoder can still encode and decode weights even in such a challenging setting, although there is an expected drop of performance.
>
> (**2b**). We next evaluate if hyper-representations can be trained on diverse zoos. For this experiment, we train a hyper-representation on the train split of zoo 3. With this, we aim to show that training hyper-representations on diverse zoos improves generalization capabilities further. **Results.** Our results show that training on diverse zoos is a much more difficult task to optimize, hence the reconstruction on the original zoo degrades. It nonetheless improves the reconstruction results on the test split of the diverse zoos 2 and 3. This indicates that varying seeds and hyperparameters may be different aspects of complexity that need to be considered.
>
> **Table 2. Generalizability towards more diverse model zoo configurations (measured as the $R^2$ reconstruction score, higher is better).**
> | Training zoo    | Test zoo 1: original | Test zoo 2: vary activation | Test zoo 3: vary hyperparameters |
> |-----------------|----------------------|------------------------------------------------------|----------------------------------|
> | Original        | 81.9%                | 45.7%                                                | 38.9%                            |
> | Diverse (zoo 3) | 25.8%                | 89.1%                                                | 75.6%                            |
>
> Overall, experiments 2a and 2b show that hyper-representation generalize beyond the zoos they were trained on, to zoos with very different distributions. Training hyper-representations on diverse zoos further improves their generalization capabilities.
>
> ## 2. Shared concern of afuw and cFnj: “high cost of collecting a model zoo”
>
> In practice, there are many existing model zoos (e.g. huggingface, pytorch pretrained models, github repos) with hundreds and thousands of models that can be leveraged by our approach at no cost. The main purpose of using the current zoos is to control the experiment design, to enable feasibility and reproducibility (training takes only ~8 hours on 24 cpu cores), detailed analysis and evaluation. With controlled experiment design, we aim to develop and evaluate inductive biases and methods to train and utilize hyper-representation, which can be scaled up efficiently later. Another interesting direction is reducing the number of required models in a zoo by training on fewer samples, but in our opinion it is out of scope for this submission.
>
> To clarify further, the original idea of hyper-representation learning ([33] in the submission) is motivated by the question if populations of neural network models can reveal more about the underlying manifold of well performing models than individual models can do. We follow this original idea of [33] in this submission in the prerequisite to have model zoos in place. We further demonstrate the cross-dataset transferability of learned representation in Sec. 4.2.3, which reduces the requirements of having a zoo for every dataset or task.

---

### Meta-Review · Area_Chair_mvrc · 2022-08-27

**Recommendation:** Accept
**Confidence:** Less certain

**Metareview:**

This paper learns hyper-representations to generate parameters of neural networks. The authors propose layer-wise loss normalization in the generation process. The proposed method is demonstrated on several tasks.

The paper received two positive reviews and one negative review.

Reviewer afuw was not convinced by the usefulness of the proposed method, and initially recommended rejection. The authors did address some key concerns of this reviewer in the rebuttal with added experiments. This reviewer raised the rating to borderline reject after reading the rebuttal, but remained concerned with relatively simple architectures and relatively small datasets. The authors further explained that this is currently the norm in this line of research.

The authors addressed most of the concerns of the other two reviewers in their rebuttals.

Overall, I feel this is an interesting exploration. Meanwhile I do share the concern of Reviewer afuw.

**Award:**

No

---

### Decision · Program_Chairs · 2022-09-14

Accept